# Untangling the changing impact of non-pharmaceutical interventions and vaccination on European COVID-19 trajectories

Yong Ge [1,2,14✉], Wen-Bin Zhang [1,2,3,14], Xilin Wu [1,2,14], Corrine W. Ruktanonchai [4,14], Haiyan Liu[5], Jianghao Wang [1,2], Yongze Song [6], Mengxiao Liu[1,2], Wei Yan[7,14], Juan Yang [8,9], Eimear Cleary[10], Sarchil H. Qader[10,11], Fatumah Atuhaire[10,12], Nick W. Ruktanonchai[4], Andrew J. Tatem [10✉] & Shengjie Lai [9,10,13✉]

Non-pharmaceutical interventions (NPIs) and vaccination are two fundamental approaches for mitigating the coronavirus disease 2019 (COVID-19) pandemic. However, the real-world impact of NPIs versus vaccination, or a combination of both, on COVID-19 remains uncertain. To address this, we built a Bayesian inference model to assess the changing effect of NPIs and vaccination on reducing COVID-19 transmission, based on a large-scale dataset including epidemiological parameters, virus variants, vaccines, and climate factors in Europe from August 2020 to October 2021. We found that (1) the combined effect of NPIs and vaccination resulted in a 53% (95% confidence interval: 42–62%) reduction in reproduction number by October 2021, whereas NPIs and vaccination reduced the transmission by 35% and 38%, respectively; (2) compared with vaccination, the change of NPI effect was less sensitive to emerging variants; (3) the relative effect of NPIs declined 12% from May 2021 due to a lower stringency and the introduction of vaccination strategies. Our results demonstrate that NPIs were complementary to vaccination in an effort to reduce COVID-19 transmission, and the relaxation of NPIs might depend on vaccination rates, control targets, and vaccine effectiveness concerning extant and emerging variants.

[1] State Key Laboratory of Resources and Environmental Information System, Institute of Geographic Sciences and Natural Resources Research, Chinese Academy of Sciences, Beijing, China. [2] College of Resources and Environment, University of Academy of Sciences, Beijing, China. [3] Lancaster Environment Center, Faculty of Science and Technology, Lancaster University, Lancaster, UK. [4] Population Health Sciences, Virginia Tech, Blacksburg, VA, USA. [5] Marine Data Center, South Marine Science and Engineering Guangdong Laboratory (Zhuhai), Zhuhai, China. [6] School of Design and the Built Environment, Curtin University, Perth, Australia. [7] Respiratory Medicine Department, Peking University Third Hospital, Beijing, China. [8] School of Public Health, Fudan University, Key Laboratory of Public Health Safety, Ministry of Education, Shanghai, China. [9] Shanghai Institute of Infectious Disease and Biosecurity, Fudan University, Shanghai, China. [10] WorldPop, School of Geography and Environmental Science, University of Southampton, Southampton, UK. [11] Natural Resources Department, College of Agricultural Engineering Sciences, University of Sulaimani; Sulaimani 334, Kurdistan Region, Sulaymaniyah, Iraq. [12] School of Mathematical Sciences, University of Southampton, Southampton, UK. [13] Institute for Life Sciences, University of Southampton, Southampton, UK. [14]These authors contributed equally: Yong Ge, Wen-Bin Zhang, Xilin Wu, Corrine W. Ruktanonchai, Wei Yan. ✉email: gey@lreis.ac.cn; A.J.Tatem@soton.ac.uk; shengjie.lai@soton.ac.uk

Since the emergence of coronavirus disease 2019 (COVID-19) and global transmission from early 2020, governments worldwide have implemented a series of non-pharmaceutical interventions (NPIs), to varying extents, in an effort to reduce local transmission of severe acute respiratory syndrome coronavirus 2 (SARS-CoV-2)[1]. The impact of these NPI strategies has been well documented[2–4], yet the consequences of long-term social restrictions have raised concerns about the potential for economic recession[5] and unintended adverse mental-health outcomes[6]. COVID-19 vaccines, which protect against infections and severe illness, provide the potential for relaxing NPIs and addressing associated economic and social burdens. However, evidence on the real-world effect of the coordinated implementation of NPIs alongside mass vaccination campaigns is still unclear. As vaccination rates increase in more countries worldwide, the impact of combined vaccination and NPI strategies over time, within the context of emerging or extant variants and their transmission capacity, should be quantified. This information is vital for informing policymakers who wish to promote public health while also easing the burden of invasive and restrictive NPIs.

Following approval of the Pfizer vaccine by the United Kingdom (UK) on 2 December 2020, mass-vaccination campaigns have commenced in countries worldwide[7]. The rollout of this vaccine, along with other COVID-19 vaccine products, such as Moderna, Johnson & Johnson, AstraZeneca, Sinopharm.Beijing, and Sputnik.V[8], has been observed to have varied efficacy against the transmission of SARS-CoV-2[9]. However, the number of confirmed new COVID-19 cases across the world remained high in 2021, and subsequent waves of transmission have occurred with emergence of more transmissible variants of concern (VOCs), e.g. Alpha and Delta due to immune evasion[10,11] and the potential for reinfection amongst previously infected or vaccinated populations[12,13]. For example, Europe reported a 7% increase in new weekly cases and 11% increase in COVID-19 attributed deaths during the week of 4 to 10 October 2021, compared with the previous week[14], despite 59.6% of the population in the European Region having been fully vaccinated by mid-September 2021[7]. Although countries in Europe have implemented various roadmaps to relax NPIs with the increase in vaccination rates since June 2021[15], rushed relaxation of NPIs could bring a risk of COVID-19 resurgence due to variation in the protection of vaccines for preventing transmission of VOCs[16].

Previous modelling studies[17–20] have preliminarily explored the implementation and effect of NPIs in the COVID-19 vaccination era. For instance, NPIs were estimated to have a higher impact in preventing infection than vaccination alone, assuming a variety of immunisation rates, during the first phase of the vaccination campaign in Italy in January 2021[19]. Furthermore, using a mathematical model informed by age structure in the UK, even under an optimistic scenario of vaccines preventing 85% of infections regardless of variant, the reproduction number was still estimated as 1.58 (suggesting sustainable transmission) after full vaccination of the population in the absence of NPIs[20]. These results suggest that NPIs should be continually implemented during mass-vaccination programmes to prevent COVID-19 transmission. In addition, levels of population immunity, which may or may not be reflective of real population level immunity, were directly defined under various scenarios in models and the geography of emerging variants and resulting vaccine effectiveness were often absent or incomplete. The gap between the de facto vaccination-immune and the vaccinated population can also lead to a misunderstanding of simulation results[16], where much uncertainty is introduced by varying vaccine efficacy and variant emergence[21]. Therefore, the real-world effect of integrated NPI and vaccination strategies is still

unclear, leading to uncertainty as to which policies and interventions are most appropriate.

In this study, we estimated the real-world impact of vaccination programmes and NPI strategies in mitigating COVID-19 transmission among populations over time, against different emerging variants, and amongst various settings. Based on large-scale and near real-time datasets, including epidemiological parameters, virus variants, vaccines, and control variables, we proposed a data-driven approach for quantifying the changes in COVID-19 transmission across 31 member states of the World Health Organization (WHO) European Region (Fig. 1a)[22], as a result of one, or a combination of both of these intervention strategies from 1 August 2020 to 25 October 2021. Given the spatiotemporal heterogeneity in COVID-19 transmission, the instantaneous reproduction number ($R_t$) was derived to represent transmissibility under government interventions and vaccination, and the instantaneous basic reproduction number ($R_{0,t}$) was used to represent intrinsic transmissibility without NPIs and/or vaccination that governments have taken to tackle COVID-19 (see Supplementary Information). The relative effect of NPIs and vaccination, thus, was defined as the contributed percentage reductions in reproduction number from $R_{0,t}$ to $R_t$, denoted as $\triangle R_t(\%) = 1 - R_t/R_{0,t}$.

As different COVID-19 vaccine products have been used across countries, the documented vaccination rate of each country was further standardised as practical vaccination rate, based on the efficacy of vaccines against SARS-CoV-2 strains circulating prior to the discovery of VOCs. The practical vaccination rate represents the fraction of populations who might have sufficient antibodies to prevent COVID-19 infection at individual level. Our model included this adjusted rate as an independent variable, to account for the reduction in $R_{0,t}$ attributed to vaccination alone. As countries implemented diverse NPI packages without coordination[23], we used an integrated stringency index of government interventions generated by the Oxford Covid-19 Government Response Tracker (OxCGRT)[24], as a proxy to estimate the general restraint of 'lockdown style' NPIs. A Bayesian model was used to evaluate the effect of NPIs and vaccination across months for each country, and then a meta-analysis for pooling national results across 31 countries was conducted to assess the overall impact of interventions on COVID-19 in the region over time. The details of the methodology can be found in Methods and Supplementary Information.

## Results

**Overall effect of NPIs and vaccination over time.** The reductions in $R_{0,t}$ contributed by NPIs gradually increased from 31% (95% confidence interval [CI]: 25–36%) in August 2020, to 44% (95% CI: 38–49%) in December 2020 when countries started mass vaccination (Figs. 1 and 2). Thereafter, the relative effect of NPIs on reducing COVID-19 transmission stabilised at about 44–47% until the practical vaccination rate exceeded 30% in July 2021. However, NPI's effect gradually dropped to 35% (95% CI: 26–43%) by 25 October 2021, with the practical vaccination rate reaching 53% in Europe. In contrast, the effect of vaccination successively increased, reaching 38% (95% CI: 30–47%) by 25 October 2021. We found that the relative importance of vaccination on flattening the trajectory of COVID-19 transmission has exceeded that of NPIs within the WHO European region since August 2021. Although the role of vaccines had been flourishing since its rollout, we saw little growth in the effect of vaccination in September - October 2021 during the circulation of the VOC Delta (Fig. 2).

As of 25 October 2021, NPI measures coupled with vaccination rates resulted in a total reduction in $R_{0,t}$ by 53% (95% CI: 42–62%), regardless of variants in circulation (Fig. 2). As the

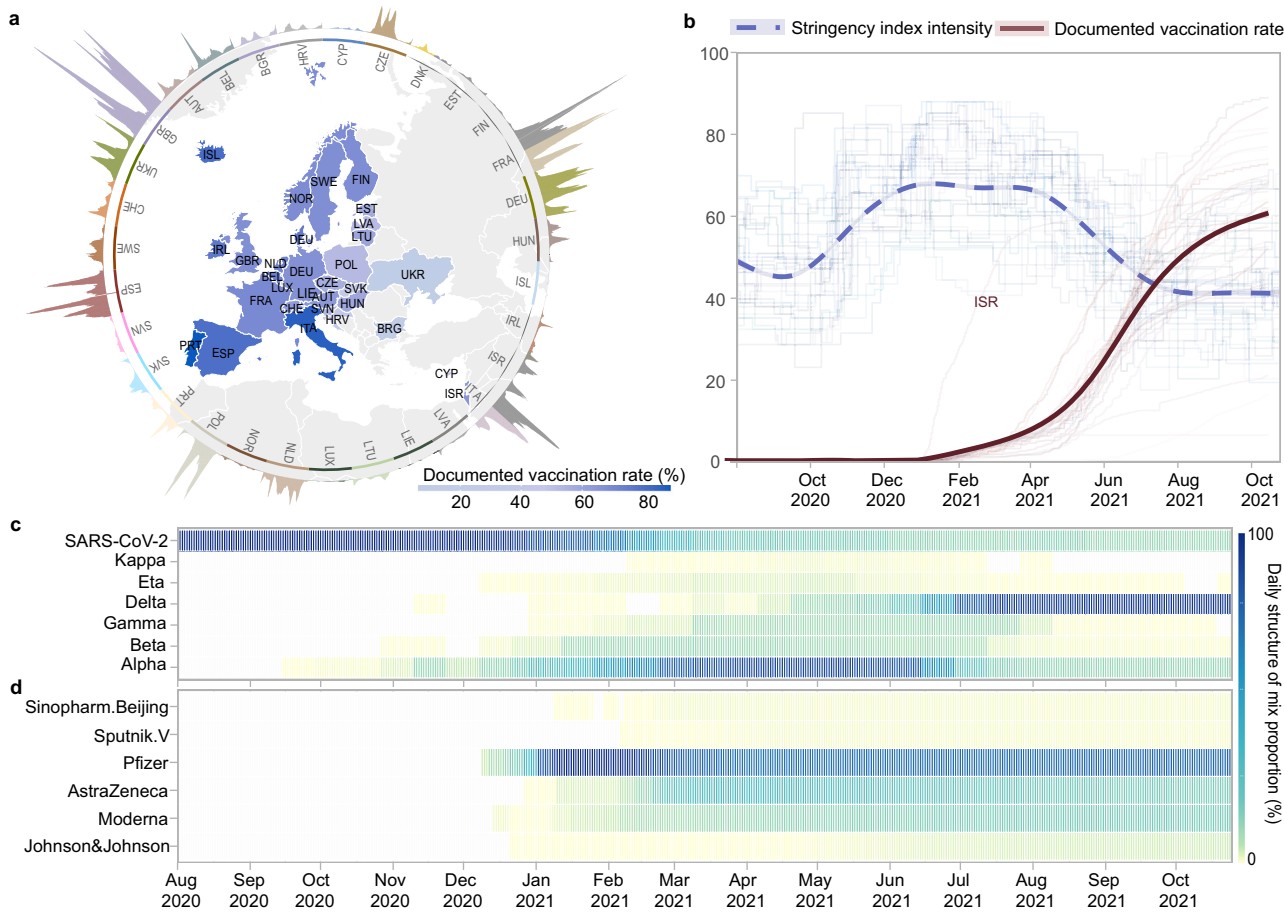

**Fig. 1 Overview of the data context in 31 countries from 1 August 2020 to 25 October 2021. a** Daily confirmed cases (outside the circle) and documented vaccination rates (inside the circle). **b** The stringency index of 'lockdown' style NPIs (shallow blue lines) and the documented vaccination rate (shallow red lines) across 31 countries. The documented vaccination rate refers to the proportion of the total population who were fully vaccinated in each country. The corresponding curves (thick blue and red lines) were fitted by the locally weighted smoothing method using national data, representing overall NPIs and vaccination rate in Europe (including Israel). **c** Daily proportion of infections caused by SARS-CoV-2 and its variants, and (**d**) daily proportion of different COVID-19 vaccine products used, where the values of each indicator within each day add up to 1. **c, d** share the same right colour legend.

evolution of COVID-19 is a complex process, and NPIs and vaccination were not the only explanatory variables for the observed reduction in $R_{0,t}$, we further used air temperature to explore the impact of environmental factors on COVID-19 transmission. During our study period, however, temperature seems to have had a small influence (0% to 3%) on reducing $R_{0,t}$ (Supplementary Fig. 2). Moreover, we also estimated reductions in $R_{0,t}$ contributed by other unknown factors (represented by residuals), e.g., personal hygiene behaviour, which were relatively stable from August 2020 to July 2021, ranging from 5% to 8%. After that, with the significantly improved vaccination rate, the unexplained reductions in $R_{0,t}$ dropped to 2% in October 2021. Our estimates for each country over time can be found in Supplementary Fig. 4.

**Interaction between vaccination and NPI effects**. As vaccinated individuals might also be protected by NPIs, we further estimated the interaction effect of NPIs and vaccination on reducing COVID-19 transmission among populations. Under a practical vaccination rate between 20% and 30%, vaccines reduced $R_{0,t}$ in populations by a median of 18%, while NPIs alone could reduce $R_{0,t}$ by 40% during the same period (Fig. 3a). However, when the practical vaccination rate reached 40–50%, the effect of vaccination (28%) surpassed that of NPIs (25%). Figure 3b showed that the overall effect of NPIs under various stringency levels tended

to decline over the study period. When the practical vaccination rate exceeded 30%, NPIs with similar stringency appeared to have a less impact on COVID-19 transmission. Furthermore, we found a gradual increase in the interaction of NPIs and vaccination for reducing $R_{0,t}$ (Fig. 3c). However, due to the low vaccination rates, the interaction effect from December 2020 to May 2021 remained small (0–6%), but it increased to 15% (95% CI: 10–19%) in September - October 2021, even against the more transmissible Delta variant. In addition, the overall effect of vaccination on preventing population-wide COVID-19 transmission increased approximately linearly with the increasing vaccination rate, whether or not the interaction effect was considered (Fig. 3d).

We also validated our model by using leave-one-out cross-validation approach for each of the 31 study countries (Supplementary Information C4), where the median of root-mean-square error (RMSE) was 0.26 (IQR: 0.24 – 0.33), and R-squared ranged from 0.35 (Poland with a vaccination rate of 52%) to 0.76 (Ukraine with a vaccination rate of 16%). Sensitivity analyses were further conducted by altering model settings and parameters to assess the robustness of model. The output showed that the overall trends in estimates were highly consistent across experimental conditions (Supplementary Fig. 13).

**Potential relaxation of NPIs amid vaccination**. Under a fixed parameter of vaccination rate on 25 October 2021, we

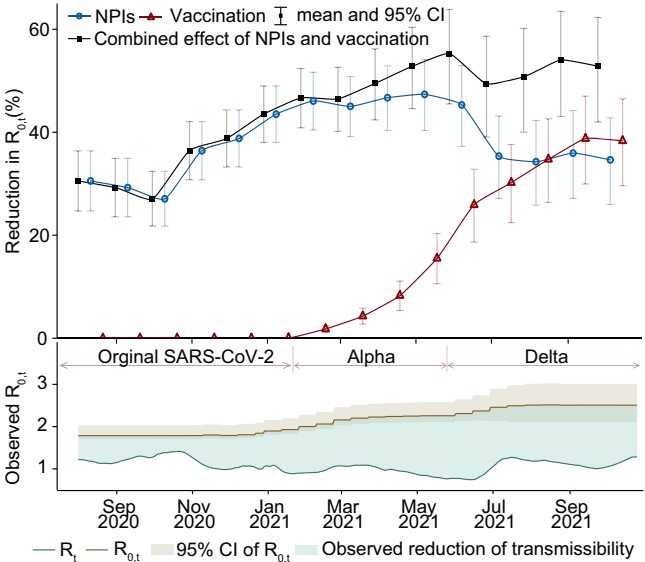

**Fig. 2 The effects of NPIs and vaccination on reducing COVID-19 transmission in Europe over time.** The overall monthly effects of interventions on reducing $R_{0,t}$ across 31 countries from 1 August 2020 to 25 October 2021 are presented with mean and 95% CI, which was pooled from national level to regional level using meta-analysis. The total effect of NPIs presented here is the effect of NPIs alone plus their interaction effect with vaccination, and the total effect of vaccination shown is the impact of vaccination alone plus its interaction effect with NPIs. In the bottom panel, the light blue area between $R_{0,t}$ (instantaneous basic reproduction number) and $R_t$ (instantaneous reproduction number [solid line]) illustrates the observed reduction of COVID-19 transmissibility. $R_{0,t}$ are presented with mean (dash line) and 95% CI (grey area). Periods in which Alpha and Delta variants were dominant (>50%) are also shown by pink lines and relevant text.

investigated the change in the stringency of NPIs required to halt COVID-19 transmission across the 31 study countries (Fig. 4a). We compared the actual NPI setting on 25 October 2021 with the scenario of $R_t$ equal to 1, to estimate the recommended increase (when $R_t$>1) or potential relaxation (when $R_t$<1) of NPI stringency. We found that most countries should maintain higher stringency of NPIs compared with their previous levels, until sufficient immunity has been acquired in the population to contain the spread of COVID-19 with $R_t$<1. For example, under the circumstance of vaccination and COVID-19 transmission by 25 October 2021, Slovakia might need to increase their NPI stringency index from 29 to 44. NPIs with a stringency index of 29 in Slovakia were generally at a moderate level, such as recommending to close school, and restrictions on gatherings between 11–100 people. Looking back at Slovakia's NPIs implemented during previous waves, in order to reach a stringency index of 44, Slovakia might need to take additional measures, such as restricting gatherings to 10 people or less. In order to provide more reliable evidence for decision-making, we further compared the requirement for changes in NPI stringency index with the output of an indicator of openness risk, modified from the OxCGRT's approach[25]. The openness risk is a case-evidenced index of risk rating, related to whether a country is ready to adopt an 'open' policy (remove/reduce NPI measures). Figure 4b shows that findings from these two indexes are generally consistent. For instance, countries falling in Group 1 should consider delaying relaxation or boosting their NPIs, and countries of Group 2 could consider relaxing their NPIs.

## Discussion

We used a data-driven approach to estimate the respective impact of NPIs and vaccination on COVID-19 mitigations among populations in Europe by 25 October 2021. We found that the effect of NPIs alone on preventing COVID-19 decreased in 2021, along with the progress of vaccine rollout and the relaxation of NPIs. The effect of vaccination on reducing population-wide COVID-19 transmission gradually increased, and surpassed that of NPIs since August 2021 (Fig. 2). However, in the context of circulation of more transmissible variants, e.g., VOC Delta and Omicron[26,27], NPIs might remain an important complementary to vaccination in reducing COVID-19 transmission before herd immunity has been reached. Our findings and approaches can potentially be used to support prompt COVID-19 mitigation policy decisions and to inform the implementation of interventions across different settings in current and future waves caused by different variants under varying vaccination rates.

The effects of NPIs and vaccination were highly correlated with the intensity of implementation and the actual vaccination coverage among populations, respectively. The higher coverage of effective vaccines indicates a larger proportion of the population with immunity against SARS-CoV-2, resulting in greater impact of vaccination on reducing the spread of COVID-19 in communities. Additionally, more stringent NPIs, such as contact reductions and travel restrictions, could further increase their effects to decrease the transmission risk of the virus. However, it is important to note that NPIs and vaccination affect the pandemic through distinct mechanisms[28]. The former physically reduces population contact and virus transmission, and the latter decreases susceptible populations by boosting immunity. Based on observations of interventions and COVID-19 trajectories, the overall impact of NPIs was found to decrease in subsequent waves, from reducing $R_{0,t}$ by 77–82% in the first wave[29,30] and 66% in the second wave in Europe[31], to 22% in October 2021 found in this study. The reduced stringency, due perhaps to policy relaxation and fatigue over time[32], would result in the decreased effects of NPIs. However, if people were fully vaccinated, they might have immunity to prevent infections, whether they adopted NPIs or not. Therefore, this might explain that the relative importance of NPIs with the same stringency also reduced when vaccination rates increased (Fig. 3b). It is also worth noting that NPIs interacted with vaccination were observed to have a similar impact on the reduction of $R_{0,t}$ of the Delta variant as they had in the period of Alpha variant circulation.

Theoretically, with the progress of vaccination campaigns targeting herd immunity, unvaccinated people in communities are also likely to be indirectly protected by vaccines. However, our study findings from empirical data demonstrated that vaccination alone might not be enough to prevent transmission of COVID-19, for the time being, in the absence of NPIs[33,34]. The real-world effects of vaccination on reducing COVID-19 transmission were observed to be lower than the efficacy of vaccines reported from the clinic trails[9]. Therefore, relaxing NPIs before attaining adequate vaccine coverage would result in a greater number of infections than would occur if NPIs were to be maintained or increased[35]. These changes could lead to a faster and larger accumulation of infections that greatly undermine the impact of vaccination efforts[36,37]. Furthermore, limited by the weakened effect of various vaccine products against different variants and the delays of vaccine development and distributions, achieving herd immunity may be a big challenge, particularly in the face of highly transmissible variants such as Omicron or even Delmicron[38] (Supplementary Information D3).

There are several limitations in this study. First, as we focused on the effect of NPIs and vaccination in preventing COVID-19 transmission, this study did not investigate the impact of these

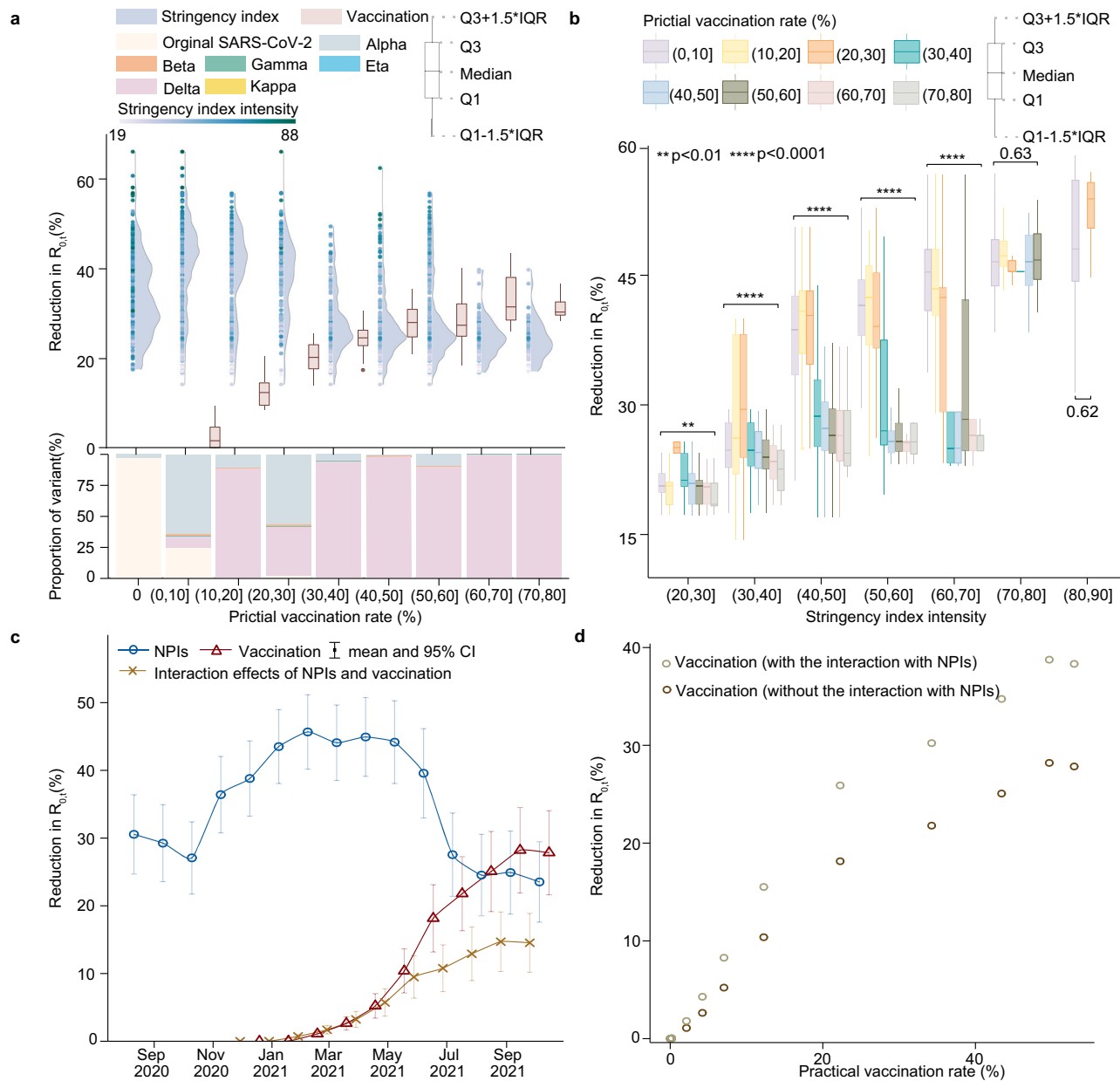

**Fig. 3 The interaction effect of NPIs and vaccination on reducing COVID-19 transmission in populations across 31 countries. a** The effects attributed to NPIs (raincloud plots in blue) and vaccination (boxplots in pink) under different practical vaccination rates. The raincloud plot visualises the intensity of stringency index (points) and the probability density of its effect. The boxplot presents the median and interquartile range. The stacked bar chart in the bottom illustrates the composition of COVID-19 variants under various vaccination levels from 1 August 2020 to 25 October 2021. The numbers of independent samples for boxplot from left (0–10% practical vaccination rate) to right (70–80% practical vaccination rate) are $n = 4434$, 962, 807, 900, 838, 1086, 497, and 187, respectively. **b** The effects of vaccines under different vaccination rates and stringency of NPIs. The effect of different practical vaccination rates within each NPI stringency group was assessed by one-way ANOVA (**$p < 0.01$, ****$p < 0.0001$). P-values are produced by two sided Wilcoxon test. The numbers of total independent samples form left (20< stringency index < = 30) to right (80< stringency index < = 90) are $n = 466$, 1334, 2173, 2055, 1805, 1387, and 459, respectively. **c** The respective effects attributed to NPIs (in blue) and vaccination (in red), and the interaction effect between NPIs and vaccination (in yellow) over time across 31 countries. **d** The comparison of vaccination effects with/without the interaction with NPIs.

interventions to reduce severe outcomes, e.g., hospitalisations and deaths, which warrants investigation in future studies. Second, although population structure such as age seems to be a major confounder[39], we have not differentiated the NPI and vaccination effects by demographic factors, due to the limited availability of case and vaccination data by age and sex across space and time. Third, vaccines might also be administered to people who were already infected, and the effect of NPIs might be negatively reduced by policy fatigue or positively impacted by adherence to personal protective behaviours against COVID-19 infections.

However, their impact has not been analysed in this work due to a lack of relevant data. Fourth, it is still unclear to what extent changes in government interventions and vaccination rate as well as waning immunity over time[40] might have a delayed impact on COVID-19 transmission and our findings[41]. We, therefore, estimated the reductions in $R_{0,t}$ caused by interventions for each month to minimise the effect of delays, and found that our main results were stable over time. Fifth, randomised control trials cannot be performed to robustly examine causality between interventions and the reduction in COVID-19 transmission, and

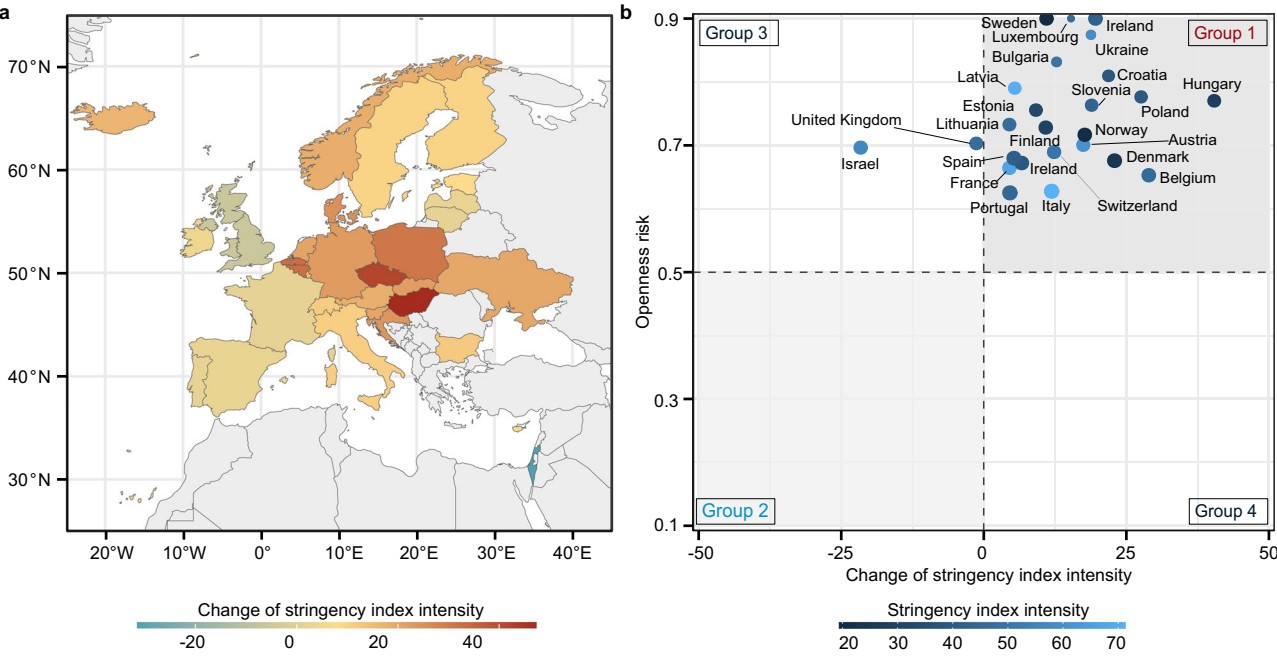

**Fig. 4 The possible relaxation of NPIs or the requirement of extra stringency to contain COVID-19 across countries. a** Under the scenario of vaccination and COVID-19 transmission by 25 October 2021, required changes of NPI stringency index to contain COVID-19 ($R_t<1$). The negative change means the degree of NPI relaxation, compared to the stringency on 25 October 2021. **b** The comparison between the estimated requirement of changes in NPI stringency index presented in (**a**) and the output of the openness risk (from 0 to 1) - an indicator modified from the OxCGRT's approach[25]. A higher openness risk ( > 0.5) means an increasing likelihood of COVID-19 resurgence, and vice versa. Countries in Group 1 (increasing NPI stringency) and Group 2 (relaxing NPIs) mean that they have consistent findings between two indicators. Groups 3 and 4 mean that the two indicators have conflicting results and extra evidence might be needed.

this study was not designed to distinguish the efficiency of individual NPIs and their interaction.

However, as with studies using similar approaches that have been conducted for countries in previous waves[28–30], empirical/observational correlations between changes of interventions and the dynamic of COVID-19 trajectories can alternatively provide important evidence for assessing the impact of interventions on reducing COVID-19 transmission. Due to the uncertainty in the duration of acquired immunity and a high likelihood of the emergence of new variants[42,43], the most effective strategy for preventing further waves of COVID-19 may include the distribution of vaccines to areas and populations with low vaccination rates, together with a certain level of combined NPIs that are more appropriate according to local socio-economic contexts and control targets.

## Methods
### Data sources and processing
*Study countries and epidemiological parameters.* The 31 study countries include Austria, Belgium, Bulgaria, Croatia, Cyprus, the Czech Republic, Denmark, Estonia, Finland, France, Germany, Hungary, Ireland, Iceland, Italy, Lithuania, Luxembourg, Latvia, Liechtenstein, Norway, Poland, Portugal, Slovakia, Slovenia, Spain, Sweden, Switzerland, Ukraine, the Netherlands, the UK, and Israel (Fig. 1a). We used the instantaneous reproduction number ($R_t$) to represent real-world COVID-19 transmission. In this study, the daily estimates of $R_t$ were obtained from the *Our World in Data* data repository and contributed by Arroyo-Marioli et al[44], with $R_t$ being estimated from the number of daily new cases using the Kalman filter[45]. Specifically, the dynamics of $R_t$ was considered as

$$R_t = R_{t-1} + \varepsilon_t, \varepsilon_t \sim i.i.d.N(0, \sigma_\varepsilon^2) \tag{1}$$

and output was defined as the growth rate ($g_t$) of COVID-19 infections, which is derived from the classic SIR model linking to $R_t$

$$g_t = \gamma(R_t - 1) + \eta_t, \eta_t \sim i.i.d.N(0, \sigma_\eta^2) \tag{2}$$

where $\gamma$ is the daily transition rate from infected to recovered, which is the inverse of the serial interval[46]. Details can be found in Supplementary Information A1. The

Kalman smoother fit $g_t$ to observed growth rates derived from empirical case data to give best estimates of $R_t$ by minimising the mean-squared error.

To derive the empirical change of transmission trend, we also estimated the instantaneous basic reproduction number ($R_{0,t}$) to capture the intrinsic transmission capability of the virus without government public health interventions. We first assembled data of the biweekly proportion of sequences of six main SARS-CoV-2 variants, including lineages B.1.1.7 (VOC *Alpha*), B.1.351 (*Beta*), P.1 (*Gamma*), B.1.617.2 (*Delta*), B.1.525 (*Eta*), and B.1.617.1 (*Kappa*), in each of the 31 study countries by 25 October 2021. SARS-CoV-2 sequence data were collected from the Global Initiative on Sharing All Influenza Data (GISAID)[47], as of 25 October 2021. According to the transmissibility of each variant, we then calculated a weighted average of basic reproduction numbers of the six variants mentioned above and the SARS-CoV-2 strain in circulation before VOCs became predominant (seven coronavirus variants in total) within each country,

$$R_{0,t} = \sum_{i=1}^{7} w_{i,t} R_{0,i} \tag{3}$$

where $w_{i,t}$ is the weight of the basic reproduction number of variant $i$ ($R_{0,i}$) at day $t$, calculated by the biweekly proportion of infections caused by that strain, based on sequence data. As $R_{0,i}$ of each variant was reported in comparison with the transmissibility of SARS-CoV-2 strains in the early stages of the pandemic, we first set a hyperprior over the basic reproduction number of SARS-CoV-2, and evaluated $R_{0,i}$ by multiplying the corresponding reported expansion parameters (Supplementary Table 1). More details of epidemiological data collation and analysis as well as limitations of using reproduction numbers to describe transmission context can be found in Supplementary Information.

*Stringency index of NPIs.* We used a large-scale dataset of NPIs collected and assembled by the Oxford COVID-19 Government Response Tracker (OxCGRT)[24]. The stringency index is a composite measure provided by the OxCGRT based on nine indicators, including eight containment and closure policy indicators (school closures, workplace closures, public event cancellations, gathering restrictions, public transport closures, stay-at-home orders, internal movement restrictions, and international travel controls) and one indicator of public information campaigns, scaled range from 0 (no interventions) to 100 (implementing the strictest NPIs). In order to investigate NPI effects over time after the first wave of pandemic, against different variants and among various settings, we studied NPI effects from 1 August 2020 - about two months before the emergence of Alpha variants. The details of the stringency index and more discussion on using this index can be found in Supplementary Information D2.

*Vaccination data.* We obtained daily data of fully vaccination rates by country and manufacturer from *Our World in Data*[48], as of 25 October 2021. Fully vaccination rate is defined as the fraction of the total population who received all doses prescribed by the vaccination protocol. However, countries generally provided several COVID-19 vaccine products, and different vaccines might have various efficacy against SARS-CoV-2 infections, even for the same strain. As this study aimed to understand the overall impact of vaccination on COVID-19 transmission among populations rather than the individual effectiveness of vaccine products with different coverages in populations, we created an indicator, practical vaccination rate, in order to account for the difference between the efficacy of vaccine products used across different countries. The practical vaccination rate for country $c$ at day $t$ was defined as:

$$V_t^c = \text{fully vaccination rate}_t^c \sum_{i=1}^{6} e_i p_{i,t}^c \qquad (4)$$

where $e_i$ was the effect of vaccine $i$ against SARS-CoV-2 circulated in the early stages of the pandemic, estimated in clinical trials (see Supplementary Table 3), and $p_{i,t}^c$ was the proportion of the vaccine products $i$ used in country $c$ at day $t$. Thus, this practical vaccination rate was calculated as a baseline to represent the population that might be directly protected by all vaccines distributed in each country.

*Index of openness risk.* The index of openness risk is based on recommendations set out by the WHO, regarding measures that should be put in place before COVID-19 response policies can be safely relaxed. The OxCGRT created this index to provide a cross-national overview of the risk and response of different countries as they tighten or and relax physical distancing measures[25]. The index of openness risk calculates a measure of probability that a country faces from adopting an 'open' policy stance (i.e., no extra NPI measures are implemented to contain the virus). We revised this indicator by further considering the vaccination rate on 25 October 2021, as a comparison with our estimated potential of NPI relaxation (Supplementary Information A4). This index ranges from 0 (lowest risk - lifting all NPIs) to 1 (highest risk - strengthening NPIs), with 0.5 being assigned as the divide between the low and high risk of openness.

*Control variables.* We also used climate data to account for seasonal and weather effects on virus activity and human behaviour, which might significantly influence COVID-19 trajectories. Daily air temperature and humidity were assembled for all 31 countries, derived from the Global Land Data Assimilation System[49]. However, only temperature was included as a control variable in our model, as humidity was found to be highly collinear with temperature during preliminary analyses, see Supplementary Fig. 12. In the modelling, all explanatory variables were normalised by min-max normalisation, ranging from 0 to 1.

**Assessing the effects of NPIs and vaccination.** We used a bottom-up approach (described in Fig. 5) to evaluate the effect of NPIs and vaccination in Europe by pooling the national effect through meta-analysis. For each country $c$, we fitted a

Bayesian model by assuming that the effect of NPIs and vaccination on reducing COVID-19 transmission was relatively stable and constant in each month $l$. We measured the empirical change from the instantaneous basic reproduction number ($R_{0,t}^c$) to the instantaneous reproduction number ($R_t^c$) as the outcome variable, representing the daily amount of the reductions in COVID-19 transmissibility against different variants, NPIs and vaccination settings over time. For each country, we built the following generalised linear model to use the variation of NPIs and vaccination explaining the reductions in $R_t$ over time.

$$R_t^c \sim \text{gamma}\left(\Phi_{t,l}^c, 0.5\right)$$
$$\Phi_{t,l}^c = R_{0,t}^c \exp\left(-\alpha_l^c N_t^c - \beta_l^c V_t^c - \lambda_l^c N_t^c V_t^c - \varphi_l^c T_t^c - \Delta_l^c\right) \qquad (5)$$

where $N_t^c$, $V_t^c$, and $T_t^c$ are the stringency index of NPIs, practical vaccination rate, and air temperature for country $c$ in month $l$ at day $t$, respectively. In addition to NPIs and vaccination, we also modelled their interaction of reducing $R_t$ by directly incorporating a product term ($N_t^c V_t^c$) in our model. Moreover, the unobserved confounders of the change between $R_{0,t}^c$ and $R_t^c$ were represented by the residual $\Delta_l^c$. To estimate the model parameters, we used a Bayesian framework to provide the estimates with prior knowledge. We assumed that $R_t^c \sim \text{gamma}\left(\Phi_{t,l}^c, 0.5\right)$. As NPIs and vaccination were likely to positively impact the trajectories of COVID-19, i.e., reducing $R_{0,t}$, we put a gamma prior with hyperprior over the coefficients of NPIs, vaccination and their interaction term in our model. Specifically, $\alpha_l^c$, $\beta_l^c$ and $\lambda_l^c$, following gamma$(u, 1)$ and $u \sim$ uniform$(0, 1)$, varied by country according to their data contexts. Additionally, we had a Gaussian prior over the coefficients $\varphi_l^c \sim N(0, k)$ and $\Delta_l^c \sim N(0, k)$, $k \sim$ Half normal$(0, 0.3)$, as temperature and other unknown factors might also be related to the transmission dynamics of the disease[50,51]. The posterior estimates of coefficients in Eq. (5) can be found in Supplementary Information B3. Finally, the relative effect of NPIs and vaccination for country $c$ in month $l$ could be calculated by $1 - \exp\left(-\alpha_l^c N_t^c\right)$ and $1 - \exp\left(-\beta_l^c V_t^c\right)$, respectively, wherein $N_t^c$ and $V_t^c$ was the average value of the stringency index of NPIs and practical vaccination rate. The effect size was defined as the reduction in $R_{0,t}$ regarding $R_t$, i.e., $1 - R_t/R_{0,t}$, with the combined effect of two independent variables calculated as the sum of the estimated effects of the two variables minus the corresponding product of their effectiveness. The prior and posterior predictive estimations can be found in Supplementary Information C1.

We estimated the effect of NPIs and vaccination for every month to account for seasonal and other potential temporal effects. This process was performed using Markov Chain Monte Carlo (MCMC) methods with Rstan[52]. We ran four chains for 2000 iterations with 500 iterations of warmup and a thinning factor of one to obtain 600 posterior samples for each month and country (see Supplementary Information C2). We validated our model using a 'leave-one-out' cross-validation approach (see Supplementary Information C4). Sensitivity analyses were also performed to assess model robustness in terms of our assumptions on parameters (see Supplementary Information B5).

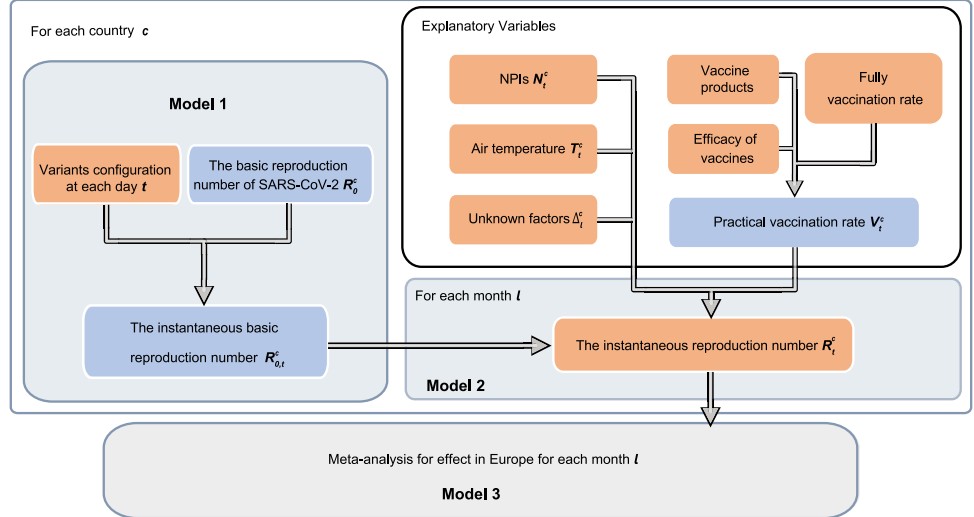

**Fig. 5 Overview of models using bottom-up approaches.** Orange nodes represent the observed data. Blue nodes represent the pseudo variables generated by the observed data. For each country, we put a prior on $R_0$ with hyperprior varying by country, where the prior mean was setted as the highest $R_t$ before 1 December 2020, see Supplementary Information A2. Then, $R_{0,t}$ representing the intrinsic transmissibility was estimated by Model 1. By comparing observed $R_t$ with $R_{0,t}$ in Model 2, we estimated coefficients of variables to assess respective effects attributed to various interventions and factors on curbing COVID-19 for each country by month. A variable, represented by the residual $\Delta$, was used to characterise the impact of other unknown factors on $R_t$ in addition to practical vaccination rate, NPIs and air temperature. Finally, the overall effect of NPIs and vaccination in the European region was evaluated in Model 3 by pooling the national effects across countries through meta-analysis with the random-effect model.

**Meta-analysis.** We pooled national effects across the 31 study countries (Supplementary Information B2) to estimate the regional effect (Figs. 2 and 3) through meta-analysis using a random-effects model[53]. Instead of directly fitting a full regional model with fixed pooling, we used the meta-analysis approach for pooling national results as it has a better performance (16% increase regarding R-squared) on explaining the variation of COVID-19 transmission across countries and higher computational efficiency in heterogeneous data context (See Supplementary Information C4). This approach also enables us to examine the effect of NPIs and vaccination on regions of interest with more flexibility. Additionally, the heterogeneity between national effects across each country was estimated using Cochran's Q and $I^2$ statistics[54]. We used a 'leave-one-out' validation to evaluate the regional results by omitting one country at meta-analysis (Supplementary Fig. 24), aiming to show the individual result effect on the overall estimate derived from the other 30 countries. All calculations were performed using the R meta package[55].

**Ethical approval.** Ethical clearance for collecting and using secondary data in this study was granted by the institutional review board of the University of Southampton (No. 61865). All data were supplied and analysed in an anonymous format, without access to personal identifying information.

**Reporting summary.** Further information on research design is available in the Nature Research Reporting Summary linked to this article.

## Data availability
The data[56] used in this study are publicly available online at https://github.com/owid/covid-19-data/tree/master/public/data. The processed climate data[57] are available online at https://github.com/wxl1379457192/Vaccine-NPIs-in-EuropeV2.

## Code availability
The modelling codes[57] for this study are available online at: https://github.com/wxl1379457192/Vaccine-NPIs-in-EuropeV2.

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

## Acknowledgements

We thank the researchers who generated and publicly shared the epidemiological, intervention and sequencing data used in this research. This study was supported by the National Natural Science Foundation for Distinguished Young Scholars of China (No. 41725006), the Bill & Melinda Gates Foundation (INV-024911), and the National Institutes of Health (R01AI160780). YG is supported by funding from the National Natural Science Foundation for Distinguished Young Scholars of China (No. 41725006). AJT is supported by funding from the Bill & Melinda Gates Foundation (OPP1106427, OPP1032350, OPP1134076, OPP1094793), the Clinton Health Access Initiative, the UK Foreign, Commonwealth and Development Office (UK-FCDO), the Wellcome Trust (106866/Z/15/Z, 204613/Z/16/Z), the National Institutes of Health (R01AI160780), and the EU H2020 (MOOD 874850). SL is supported by funding from the Bill & Melinda Gates Foundation (INV-024911) and the National Natural Science Foundation of China (81773498). JY is supported by funding from the Shanghai Municipal Science and Technology Major Project (ZD2021CY001). The funder of the study had no role in study design, data collection, data analysis, data interpretation, or writing of the report. The corresponding authors had full access to all the data in the study and had final responsibility for the decision to submit for publication. The views expressed in this article are those of the authors and do not represent any official policy.

## Author contributions

Y.G., W.B.Z., X.L.W., and S.J.L. conceived and designed the study, built the model, collected data, finalised the analysis, interpreted the findings, and wrote the manuscript. Y.Z.S. and M.X.L. collected data. C.W.R., H.Y.L., and J.H.W. interpreted the findings, and revised drafts of the manuscript. W.Y., J.Y., N.W.R., E.C., S.H.Q., F.A., and A.J.T. interpreted the findings, and commented on and revised drafts of the manuscript. All authors read and approved the final manuscript.

## Competing interests

The authors declare no competing interests.
