## [Peer Review File · Nature Communications]

Untangling the changing impact of non-pharmaceutical interventions and vaccination on European COVID-19 trajectoriesREVIEWER COMMENTS

Reviewer #1 (Remarks to the Author):

Summary

This study aims at quantifying the relative impact of the implementation of NPIs and vaccinations strategy, on temporal COVID19 transmission, using a data-driven approach. They used full vaccination data from 27 countries, the stringency index of NPIs data, COVID19 vaccines effectiveness data, and air temperature, to assess the relationship between the basic and the instantaneous reproduction number through a generalized linear model. The regional combined effectiveness of NPIs & vaccination was obtained through a meta-analysis of national effectiveness, using a mixed modeling framework.

This study is original in the sense that it utilizes a data-based approach whilst studies on similar topics mostly use a mathematical modeling approach. In a time when countries are battling new COVID19 variants, and public health authorities need to make decisions on whether or not to completely remove or partially ease non-pharmaceutical interventions, this study provides insights on the combined effect of NPIs & vaccination.

Overall, my minor concerns with the paper (all of which could potentially be dealt with at revision) are:

1. The authors state that only those that fully vaccination rate is the statistic of the fraction of the population who received at least 2 doses of the COVID19 vaccine. However, given that they included the Johnson & Johnson vaccine in their study, this is either a mistake in this definition or an intrinsic problem in the calculation, if this was coded as mentioned.

2. In the 'Supplementary Information', the authors discuss at length the limitation of R_0 , based on the various methods that could be used to calculate it and the variability of the resulting estimates. If I understand correctly, the method used to estimate the basic reproduction number R_0 , is based on the assumption that all identified cases were found through mass testing. This allowed the authors to derive R_0 from a SIR model rather than an SEIR one. It is unclear whether the relaxation of this assumption would have a significant impact on the results as currently presented. More information on the evidence that led the author to make such a strong assumption would be helpful. Alternatively, a discussion about how their results would change (if they do) if they assume that most cases might have been identified from symptomatic patients and their contacts, rather than mass testing.

Lastly, I would like to state that I really like figure 1. It is a bit heavy on information, but it is useful to get an overview of all included data, combined into a single figure.

Reviewer #2 (Remarks to the Author):

- There are several good things about this paper. The research question is important. The authors collected some data, which is always nice. The writing is overall OK, the Figures are nice. I will now go through my concerns:

- 1. Model improvements:

- A combination of high model complexity (different degrees of pooling at various parts of the model), missing information, and confusing presentation, makes it hard to understand the model (see further below). But here is how I understand the model:

- As far as I understood it, you don't use an end-to-end model from inputs to cases. Instead you estimate R_t from cases in a one model (model 1), compute R_{0t} from a prior and the variant composition at each point in time in another, separate model (model 2), and then use R_t over $R_{0,t}$

as the regression target in your main model (model 3). Model 3 is fitted separately for each country (no pooling) and each month, and results are averaged across countries.

- If my understanding is correct, then there are probably several ways in which the model could and should be improved:

- 1a) Model interaction between vaccines and NPIs directly: The key of the paper is the interaction between NPIs and vaccines. However, if we look at the main equation (between lines 448 and 449), we see that NPI effects and vaccination effects are models as independent. Why not model the interaction directly, rather than going via this convoluted route of splitting the model up by months.

- 1b) Use partial pooling: It seems that you should use partial pooling (maybe even full pooling) across countries. Currently, you use no pooling across countries, which runs into obvious failure modes. E.g. assume that some few countries have high vaccination rates and many other countries have very low vaccination rates. A pooled model can deal with this just fine. However, in your model, there is no data to identify beta in the countries with low vaccination rates, and the posterior over beta will be close to the prior. In the end, you average the posterior of the countries with and without vaccination, and you get a posterior that's close to the prior.

- 1c) Improve model of $R_{0,t}$: For estimating relative differences between the effectiveness of NPIs and vaccines, the estimate for $R_{t,0}$ does not have to be perfect. However, the paper gives absolute reductions in R_t from NPIs and vaccines. Where does the model get the evidence to estimate absolute reductions in R_t ? This evidence comes mainly from the comparison of R_t to $R_{t,0}$. So the model of $R_{t,0}$ should be really good. The appendix shows that the authors take a basic reproduction number from before the start of the first wave (multiplied by factors for the current variant composition). So the authors implicitly assume that this basic reproduction number (from before/the start of the first wave) would still give you the reproduction number in the latter half of 2020 if there were no NPIs and vaccinations in place. This is, of course, totally wrong, because the population behaviour changed massively, and with it the basic reproduction number. Additionally, the 95% credible Interval on R_0 that is given in the results is 3.02 - 3.35, which is way too small, considering the large inter-country variation in R_t . One possible solution would be to fit a full Bayesian model with a hyperprior over R_0 (and let it vary by country).

- 2. Communication improvements, adding missing information, model verification :

- It is difficult to understand the model and assess the suitability of the used model. A combination of high model complexity (different degrees of pooling at various parts of the model), missing information, and confusing presentation, makes it hard to understand the model. How to improve:

- ****Improve presentation****:

- 2a) include a model graph that highlight the various sub-models and the degree of pooling used (see above).

- 2b) Also, the model is currently partially explained in different places. E.g. the fact that you fit a separate model for each country and month is only explained in the results, not in the methods. I recommend focussing on writing a good, concise (!), and comprehensive model description in the methods.

- 2c) ****Correct mistakes and remove unclarities****:

- Take, for example, the equation between line 448 and 449:

- First, this equation is wrong, as you say yourself in line 453. In fact, there is output noise on R_t , which is not included in this equation.

- also, I assume that x , y , and z should likely be x_t , y_t , and z_t . Issues like this make it harder to understand the model.

- What about epsilon? Is there a different epsilon for each day (i.e. you should write ϵ_t), or

only one per month? This is pretty crucial for understanding the model

- ****Add missing information:****

- 2d) This is a Bayesian analysis, but most priors are not given. E.g. what were the priors for alpha and beta?

- 2e) Especially, what was the prior for epsilon and how was it chosen? The prior on epsilon determines how much of the changes in R_t the model will explain with NPIs/vaccines, and how much it will attribute as to noise.

- (if you don't have already, you should probably have a hyperprior over epsilon)

- ****Add missing analysis for model verification****

- 2f) There are no prior or posterior predictive checks. I expect these in every Bayesian modelling paper (in the Supplement). How am I supposed to evaluate your model if I can't see the model fit?

- 2g) Ideally, you'd also do leave-one-out cross validation (or other forms of cross-validation) where you show the predictions of your model on unseen data. I don't think that the NPI stringency index and the "practical vaccination rate" alone can explain much of the changes in R_t , so I expect most of the fit to be carried by epsilon, and accordingly I expect the model to have very little predictive power.

- 2h) Epsilon is pretty crucial in your analysis. Please show epsilon over time (together with alpha and beta over time). Also show gamma.

- ****Other missing information****

- 2i) Show the results for coefficients: All plots that claim to show effectiveness show $1 - \exp(-\alpha_i * x)$. In my mind, the effectiveness of NPIs and vaccines is only $1 - \exp(-\alpha)$, while x represent how strict/high the NPIs/vaccination rates were. $1 - \exp(-\alpha_i * x)$ should not be called "effectiveness", but rather something like "realised reduction in R_t ", or so. But regardless of terminology, you should show alpha and beta over time. The abstracts makes strong claims that you show how the effectiveness of NPIs develops as vaccination rates increase. But, in fact, Figure 2 might only show that fewer NPIs were used as vaccination rates increase.

- 2j) Your dataset on vaccine efficiency by subtype: It is great that you collected this dataset. But please include some information about how you collected it. Also, Table A2, which shows the dataset, has no citations/sources what-o-ever. That is very suboptimal if the dataset is supposed to be a contribution.

- 3) specific issues:

- 3a) why negative binomial distribution on R_t (line 453)? R_t is a continuous variable, the negative binomial distribution is a discrete distribution.

- 3b) The English is not perfect, but understandable. Even abstract has sentences that I don't understand, e.g. The effectiveness of NPIs alone declined approximately 23% since the introduction of vaccination strategies, where the relaxation of NPIs promoted the decline from May 2021. There are also some incorrect words used here and there. The paper would benefit from language editing

- 3c) How did you select the countries?

- 3d) You use the "national daily practical vaccination rate" as this input variable. This is a misnomer, as this number is already calculated from vaccine efficiency numbers, not only vaccination rates. But

besides, this number should already pretty much give you the reduction in R_t from vaccines at each point in time, so it's curious to use this as the model input. At the very least, you should compare the "national daily practical vaccination rate" against the reduction in R_t estimated by your model, and show if they are similar or not (and if not, explain why not)

- 3e) question on the results: how is it possible that e.g. both NPIs and vaccines reduce R by $\sim 25\%$ in Sept 2021, but R overall is reduced by $> 75\%$ (see figure 2 bottom)

- In general, why are these reductions so small? R was mostly around 1 in these periods, how did that happen if not through NPIs and vaccines? Probably epsilon in your model?

- 3f) The vaccination effects that you get, even when factoring in that not the whole population is vaccinated, seem pretty inconsistent with results from clinical trials (which show higher effectiveness). Do you have any idea why?

- 4) minor points:

- abstract should give countries and time window of analysis

- Quote: "However, clinical trials estimating vaccine efficacy 61 were conducted when novel variants, such as the Delta-variant, had not yet emerged7,"

- not longer true now

- the intro could be shortened. I did not count words, but it feels long.

- how are explanatory variables normalised? what is the range e.g. of stringency index

Reviewer #3 (Remarks to the Author):

The authors used an epidemiological model to estimate empirical effectiveness of NPIs in the presence of vaccination and how vaccines affect the effectiveness of NPIs in 27 European countries. They found that NPIs are still important to suppress the virus transmission despite the administrations of vaccines in these European countries, which agrees with other recently published work (Sonabend R. 2021; Leung K. 2021). They also found that the administration of vaccines may reduce the effects of NPIs on reducing the virus transmission, while further clarifications about the methods are needed to assess this conclusion.

One advantage of this work is the way that the authors dealt with the heterogeneous transmissibility of the changing viral variants and various effectiveness of different vaccines across different countries. This could be helpful for future work that hopes to synthesise evidence from such complex real-world situations. However, I would be particularly interested to know why the effectiveness of NPIs and vaccinations look almost identical to the observed stringency index of NPIs and vaccination rate. This is critical to interpret the findings as well.

The authors studied an important and timely question, which could potentially be useful in informing the policy making. However, the fact that the authors used the stringency index to measure NPIs may limit the usefulness of their findings, as the stringency index is not very interpretable and people are still unsure about which NPIs should be implemented or relaxed. In addition, the authors adopted transmission (measured by reproduction number) as outcome instead of disease burden (i.e., severe outcomes after infection), which may further limit the implications of their findings as the disease burden may be substantially reduced by the vaccines while the virus is still transmitting. These limitations should also be discussed when interpreting their findings.

The writing of the manuscript could be further improved. Particularly, figures need to be appropriately referenced, while more details are needed in the methods. In addition, statements in the introduction and discussion could be more cautious.

Specific comments:

* I would suggest the authors to include page number and line number in the manuscript, so that it would be easier to locate.

Introduction

* "By mid-September 2021, the vaccination rate had reached 59.6% ...": Are these numbers for partial or complete vaccination?

* "Despite these vaccination rates, a subsequent wave of Covid-19 cases emerged in July 2021 with daily confirmed cases of nearly 40,000, driven primarily by the emergence of the novel Delta variant." This statement implies the decreased effectiveness of vaccines on preventing infections from the Delta variant, which may be true. In particular, the wave can be caused by infections among unvaccinated people, therefore it needs support from references that reported the vaccination status among the confirmed cases during this July 2021 wave.

* "...as to the true threshold of herd immunity" Do the authors mean the threshold of vaccination rate to reach herd immunity?

* "...epidemiological model-based numerical simulations...": It does not read like a typical term in the field.

Results

* The description of instantaneous basic reproduction number, instantaneous reproduction number and the estimation of effectiveness of NPIs and vaccines can be further improved. Currently, it is not very clear how these key parameters were defined and estimated without further reading the methods and supplements.

* "... effectively immune via vaccination" - Do the authors mean the overall effectiveness of vaccines against variants? Perhaps need some clarification here.

* The respective effectiveness of NPIs and vaccination over time (upper panel in Figure 2) looks almost identical with the observations of these two variables in Figure 1b. Did the authors estimate the overall effect or effect per unit of NPIs and vaccinations? If overall effect, the changes in the effectiveness may just because the changes in the stringency of NPIs and vaccination rates. If effect per unit was estimated, it is interesting to investigate why the effects changed in the same way with that of the data. It is not very clear how did the authors estimate the effectiveness based on the descriptions in results and methods.

* "Before the onset of vaccination, NPIs alone controlled the practical transmissibility, measured by, to about 1.07 (1.00 - 1.15) together with the unobserved confounders but still larger than 1." - It's not clear which figure this sentence refers to and hard to follow.

* "While the combination of NPIs and vaccination has decreased a larger share of during this period than before together with the same unobserved confounders." - The sentence is confusing.

* "The stringency index of NPIs should maintain 60 currently, while it can relax to 44 in the post-vaccination era." It is hard to interoperate which NPIs should be implemented or relaxed so that we could maintain the stringency index as 60 or 44.

* "openness risk" - worth a brief introduction as it's not a very well-known term. Also need to mention this in the methods.

* "We found that our forecasted variations on NPIs implementation to stop Covid-19 were highly correlated..."

— 0.49 does not look like a very high correlation, though the test is significant.

— It is interesting to note that there are a group of countries in the bottom-left (UK, Ireland, Portugal, Spain and Iceland), which seemed very different from the rest in the top-right. Actually, predictions for the vast majority of countries in the top-right look like negatively correlated with the openness risk, which is the opposite to the reported. What are the differences for those countries in the bottom-left?

Discussion

* "Where the synergistic effect of NPIs and vaccination was 46.9% in September 2021." Not sure which plot this sentence refers to.

* "Implementing NPIs with higher strength, such as restrictions of gatherings of more than 10 people compared to that of more than 1000 people, would further decrease the potential contact population of infections. Under the circumstance, the susceptible population is harder to contact the infections and become new infections then, if the probability of getting infected after contact is unchanged. Our estimates of NPIs effectiveness are consistent with these findings, and we also provide the effectiveness of NPIs over time to counter the influence of policy fatigue." Not sure how did the authors conclude these. I don't think findings from this study support this as no individual NPIs were investigated.

* "...facing aggressive variants such as the Delta variant, over 80% of people need to have immunity to achieve herd immunity." Assuming 100% effectiveness of protection from the vaccines? It is very unlikely to be true.

* "The very population attacked by the recent outbreak of Covid-19, caused by the variants Delta, in China was children instead of previous young people [35], because most adults have been fully vaccinated. It evidenced the importance of the continued implementation of NPIs." I agree that children may become more susceptible to infection due to their lower vaccination rate compared to the adults (although fully vaccinated does not mean no infection). However, I doubt if the cited reference can actually support that the Delta variant more likely to hit children. That the Fujian outbreak hit the children was also because the outbreak happened in a school setting, which was initiated by a student who was infected by the parent. At the same time, another Delta-related outbreak in Yangzhou hit mostly old adults initially, as the outbreak was seeded in two mah-jong places.

* "..., unless they take precautions such as getting vaccinated and wearing masks." Not sure if I would fully agree with this statement. People can get infected with full vaccinations and mask wearing (Martín-Sánchez 2021). Consider to revise to something like "...due to the uncertainty of duration of protection."

Methods

* SI Table A2 - suggest to provide references for these estimates.

* First equation in "Assessing the effectiveness of NPIs and vaccinations" section is confusing.

— Does the model fit to each county separately or together? If together, need to indicate the country in the notions.

— Do the dependent variable (x, y, z) change over time and country?

— How do x and y correlate with the estimates in the previous sections (e.g., daily protection rate)?

* Justify why hard norm distribution was used.

* Suggest to provide equation for the generalised additive regression.

* "Finally, the impact of vaccination on the effectiveness of NPIs was defined by the difference between NPIs effect before and after the vaccination onset."

— Need more details of how the difference was calculated. Is it absolute additive or ratio? An equation may be helpful.

— An important assumption here is that the authors assumed all changes in effectiveness of NPIs were caused by vaccinations, while other confounders do exist (like fatigue). Need to discuss this as a limitation.

* MCMC - no description about the estimated parameters, prior distributions, likelihood functions, algorithms, iterations and diagnosis.

* Meta-analysis - It is not clear which figures are results from this analysis.

Figures

* Figure 1

— panel b - Does the "documented vaccination" indicate partial or full vaccination?

— panel c - Does each column adds up to 1? Do panel c and d share the same color legend?

* Figure 4:

— panel a - suggest to indicate the variable name and unit in the color legend.

— panel b - not very clear what are the x- and y-axis based on the legend.

References:

Leung, Kathy, Joseph T. Wu, and Gabriel M. Leung. Effects of adjusting public health, travel, and social measures during the roll-out of COVID-19 vaccination: a modelling study. *The Lancet Public Health* 6.9 (2021): e674-e682.

Martín-Sánchez, Mario, et al. COVID-19 transmission in Hong Kong despite universal masking. *Journal of Infection* (2021).

Sonabend, R, et al. Non-pharmaceutical interventions, vaccination, and the SARS-CoV-2 delta variant in England: a mathematical modelling study. *The Lancet* (2021).

POINT-BY-POINT RESPONSE TO REVIEWER COMMENTS

Reviewer #1 (Remarks to the Author):

Summary

This study aims at quantifying the relative impact of the implementation of NPIs and vaccinations strategy, on temporal COVID19 transmission, using a data-driven approach. They used full vaccination data from 27 countries, the stringency index of NPIs data, COVID19 vaccines effectiveness data, and air temperature, to assess the relationship between the basic and the instantaneous reproduction number through a generalized linear model. The regional combined effectiveness of NPIs & vaccination was obtained through a metanalysis of national effectiveness, using a mixed modeling framework.

This study is original in the sense that it utilizes a data-based approach whilst studies on similar topics mostly use a mathematical modeling approach. In a time when countries are battling new COVID19 variants, and public health authorities need to make decisions on whether or not to completely remove or partially ease non-pharmaceutical interventions, this study provides insights on the combined effect of NPIs & vaccination.

Reply: We thank the reviewer for the comments and for summarising the potential contribution and public health implication. We have made substantial attempts to address all comments below.

Overall, my minor concerns with the paper (all of which could potentially be dealt with at revision) are:

1. The authors state that only those that fully vaccination rate is the statistic of the fraction of the population who received at least 2 doses of the COVID19 vaccine. However, given that they included the Johnson & Johnson vaccine in their study, this is either a mistake in this definition or an intrinsic problem in the calculation, if this was coded as mentioned.

Reply: Thank you for pointing this out. It was a mistake in the description of the definition. The fully vaccination rate used in our study was assembled by *Our World in Data* (Mathieu et al., 2021), and people who received only one dose of Johnson & Johnson vaccine have been included in the statistic of the full vaccination rate. We have corrected the definition as “*Fully vaccination rate is defined as the fraction of the total population who received all doses prescribed by the vaccination protocol.*” in Method, Vaccination data, and “*The reported fully vaccination rates across countries were the proportion of the whole population who have received all doses prescribed by the vaccination protocol.*” in Supplementary Information A3. The relevant description in Introduction has been removed.

2. In the ‘Supplementary Information’, the authors discuss at length the limitation of R_0 , based on the various methods that could be used to calculate it and the variability of the resulting estimates. If I understand correctly, the method used to

estimate the basic reproduction number R_0 , is based on the assumption that all identified cases were found through mass testing. This allowed the authors to derive R_0 from a SIR model rather than an SEIR one. It is unclear whether the relaxation of this assumption would have a significant impact on the results as currently presented. More information on the evidence that led the author to make such a strong assumption would be helpful. Alternatively, a discussion about how their results would change (if they do) if they assume that most cases might have been identified from symptomatic patients and their contacts, rather than mass testing.

Reply: Sorry for the confusion. First, we put a prior over R_0 instead of estimating R_0 by ourselves. The assumption that all identified cases were found through mass testing was our unclear description which has been removed. In the previous version of our manuscript, R_0 was processed as a random variable, i.e., $R_0 \sim N(3.25, k)$, $k \sim \text{Half norm}(0, 0.5)$. This prior distribution referenced from Brauner et al., (2021). As there are rare studies evaluated R_0 for July to September 2020 by country, in the revised manuscript, we fit a full Bayesian model with a hyperprior over R_0 and let it vary by country to account for potential variation across countries, where $R_0 \sim \text{Gamma}(\max_{2020-08-01 < t < 2020-12-01} R_t, k)$, $k \sim \text{Half normal}(0, 0.5)$. In this way, we let the model determine R_0 according to the data context of each country. This confusion may be caused by the section title of Supplementary Information A1, **Derivation of the basic reproduction number from SIR model**, which has been revised as **Derivation of the instantaneous reproduction number from SIR model**.

Second, R_t used in this study was collected from a publicly available dataset shared in the *Our World in Data*, contributed by Arroyo-Marioli et al. (2021), and their estimates of R_t for Covid-19 were based on a structural relationship derived from the SIR model. Supplementary Information A1 only aims to give a background knowledge about how to derive such a structural relationship from the SIR model. And they don't have such a strong assumption either. The adoption of a SIR model, rather than a SEIR model, in their method was based on some considerations below. To use the SEIR model, they would have to estimate the number of currently exposed individuals. Doing so would triple the number of model parameters, such as average duration of the incubation period, relative infectiousness of exposed and infectious individuals. In simulations, they found that their estimator derived from the SIR model produces accurate estimates even when the true model is SEIR rather than SIR. We have discussed this in Supplementary Information A1.

In the revised manuscript, we clearly stated that “*In this study, the daily estimates of R_t were obtained from the Our World in Data data repository and contributed by Arroyo-Marioli et al.⁴⁴, with R_t being estimated from the number of daily new cases using the Kalman filter⁴⁵.*” in **Epidemiological parameters**. We also added the following statement at the beginning of Supplementary Information A1, “*In this study, the instantaneous reproduction number, denoted as R_t , was estimated using the method outlined by Arroyo-Marioli et al.¹. Here we give a brief description of how to derive R_t from the SIR model.*”, and at the end by “*The adoption of an SIR model, rather than a SEIR model, in the method of Arroyo-Marioli et al.¹ was based on some considerations below. To use the SEIR model, they would have to estimate the number of currently exposed individuals. Doing so would triple the number of model parameters, such as average duration of the incubation period, relative infectiousness of exposed and infectious individuals. In simulations, they found that their estimator*

derived from the SIR model produces accurate estimates even when the true model is SEIR rather than SIR.”.

Lastly, I would like to state that I really like figure 1. It is a bit heavy on information, but it is useful to get an overview of all included data, combined into a single figure.

Reply: Thanks. We hope Figure 1 could help readers to better understand the context of this study.

Reviewer #2 (Remarks to the Author):

- There are several good things about this paper. The research question is important. The authors collected some data, which is always nice. The writing is overall OK, the Figures are nice. I will now go through my concerns:

Reply: We greatly appreciate the comments and suggestions from the reviewer. We have made substantial attempts to address these comments in the revised manuscript and believe our analyses/manuscript have been significantly improved. Please find below detailed point-by-point responses to the suggestions and concerns raised by the reviewer.

- 1. Model improvements:

- A combination of high model complexity (different degrees of pooling at various parts of the model), missing information, and confusing presentation, makes it hard to understand the model (see further below). But here is how I understand the model:

- As far as I understood it, you don't use an end-to-end model from inputs to cases. Instead you estimate R_t from cases in a one model (model 1), compute $R_{0,t}$ from a prior and the variant composition at each point in time in another, separate model (model 2), and then use R_t over $R_{0,t}$ as the regression target in your main model (model 3). Model 3 is fitted separately for each country (no pooling) and each month, and results are averaged across countries.

Reply: Thank you for pointing this issue out. Your understanding of the modelling framework is partially correct. In fact, we first estimate $R_{0,t}$ from R_0 of SARS-CoV-2 at each time point for each country (model 1). Then, we use R_t over $R_{0,t}$ as the regression target in our main model (model 2). Model 2 is fitted separately for each country and each month. Finally, we pool the national results from model 2 to evaluate the regional situation by month through meta-analysis (model 3). To give a clear overview of our analyses, in this revision, we added a figure (Fig. 5) to illustrate our model in **Methods**, with a description of our modelling approach being provided: *“We used a bottom-up approach (described in Fig. 5) to evaluate the effectiveness of NPIs and vaccination in Europe by pooling the national effect through meta-analysis. For each country c , we assumed that the effectiveness of NPIs and vaccination was relatively constant in each month l and fit a Bayesian model by month to estimate the effectiveness of NPIs and vaccination.”*

Noting that the used R_t in model 2 is collected from a publicly available dataset (from *Our World in Data* by Arroyo-Marioli et al., (2021)). The methodology and formula

have been described in **Methods** of the revised manuscript. We stated that “*In this study, the daily estimates of R_t were obtained from the Our World in Data data repository and contributed by Arroyo-Marioli et al⁴⁴, with R_t being estimated from the number of daily new cases using the Kalman filter⁴⁵.*” We also revised the Supplementary Information A1 and added the following statement at the beginning of this section “*In this study, the instantaneous reproduction number, denoted as R_t , was estimated using the method outlined by Arroyo-Marioli et al.¹. Here we give a brief description of how to derive R_t from the SIR model.*”.

- If my understanding is correct, then there are probably several ways in which the model could and should be improved:

- 1a) __Model interaction between vaccines and NPIs directly__: The key of the paper is the interaction between NPIs and vaccines. However, if we look at the main equation (between lines 448 and 449), we see that NPI effects and vaccination effects are models as independent. Why not model the interaction directly, rather than going via this convoluted route of splitting the model up by months.

Reply: According to your suggestions, in this revision, we added a new variable (stringency index of NPIs multiplying the practical vaccination rate) in our model to estimate the interaction effect between vaccines and NPIs. We found that the interaction effect of NPIs and vaccination increased to 13% [95% CI: 9 - 17%] after September 2021, even facing the more transmissible Delta variant. The estimated interaction effect of NPIs and vaccination also suggests that NPIs were complementary or even synergistic to vaccination in the effort to end the COVID-19 pandemic before reaching herd immunity, at least in the short term. Besides, we splitted the model by month because the effect of NPIs or vaccination on reducing R_t might vary over time amid our study period, due to the policy fatigue, vaccination campaign, and changing behaviour. As we don't know how the NPI effect varies across time, under the assumption that the effect of NPIs might be relatively stable within a short period (i.e. a month), we splitted the model up by months. We have updated Results based on the revised model.

- 1b) __Use partial pooling__: It seems that you should use partial pooling (maybe even full pooling) across countries. Currently, you use no pooling across countries, which runs into obvious failure modes. E.g. assume that some few countries have high vaccination rates and many other countries have very low vaccination rates. A pooled model can deal with this just fine. However, in your model, there is no data to identify beta in the countries with low vaccination rates, and the posterior over beta will be close to the prior. In the end, you average the posterior of the countries with and without vaccination, and you get a posterior that's close to the prior.

Reply: Actually, we used pooling across the estimates of various effectiveness produced by each country instead of pooling across the input data for the study countries. We agree that pooling is an important issue for our analysis because the data contexts vary across countries. Your suggestion to use full pooling across countries may estimate a more robust effectiveness of vaccination, but we aim to untangle the effect of NPIs and the real-world effectiveness of vaccination in the European region. Instead of putting all the data together to fit a European model, we fit a model by country and pool the results across countries for each month by meta-

analysis. Specifically, the regional combined effectiveness of NPIs & vaccination in our study was obtained through a meta-analysis of national effectiveness, using a mixed modelling framework. We found that pooling the national effectiveness has better performance on leave-one-out cross validation compared to pooling the data across countries, see Supplementary Information C4. Moreover, we can more easily combine the national results across our interest region, which is more computational efficient than fitting a pooling model for each time. We have added these into the **Meta-analysis** section of Methods and the Supplementary Information.

- 1c) Improve model of $R_{0,t}$: For estimating relative differences between the effectiveness of NPIs and vaccines, the estimate for $R_{t,0}$ does not have to be perfect. However, the paper gives absolute reductions in R_t from NPIs and vaccines. Where does the model get the evidence to estimate absolute reductions in R_t ? This evidence comes mainly from the comparison of R_t to $R_{t,0}$. So the model of $R_{t,0}$ should be really good. The appendix shows that the authors take a basic reproduction number from before the start of the first wave (multiplied by factors for the current variant composition). So the authors implicitly assume that this basic reproduction number (from before/the start of the first wave) would still give you the reproduction number in the latter half of 2020 if there were no NPIs and vaccinations in place. This is, of course, totally wrong, because the population behaviour changed massively, and with it the basic reproduction number. Additionally, the 95% credible Interval on R_0 that is given in the results is 3.02 - 3.35, which is way too small, considering the large inter-country variation in R_t . One possible solution would be to fit a full Bayesian model with a hyperprior over R_0 (and let it vary by country).

Reply: Yes, we have revised our prior over R_0 according to your suggestion. We agree that long-term restrictions on our lifestyles would have significantly changed our intrinsic behaviours, therefore leading to a smaller R_0 compared to the R_0 estimates of initial outbreaks. In the revised manuscript, we set different prior mean over R_0 for each country, by using the highest R_t between 1 August 2020 and 31 November 2020 (i.e. the highest transmissibility of COVID-19 during the relaxation period of NPIs after the first wave). We assumed that the highest R_t could represent the basic reproduction number under the population behaviour change after the initial outbreaks and interventions. Compared to the estimates of R_0 in the first wave, the highest R_t was smaller by different extent across countries (see Table A2 below). In this way, we fit a full Bayesian model with a country-specific hyperprior over R_0 as you suggested.

We revised the description of the prior of R_0 accordingly in Supplementary Information A2 as “*Additionally, despite the existence of policy fatigue⁷, population behaviour amid our study period might have been altered by long-term NPI implementation compared to before the COVID-19. Therefore, we used the highest R_t between 1 August 2020 to 1 December 2020 before vaccines rolled out, as the prior mean over R_0 of SARS-CoV-2 for each country.*

$$R_0 \sim \text{gamma}(\max_{2020-08-01 < t < 2020-12-01} R_t, k),$$

$$k \sim \text{half_normal}(0, 0.5),$$

reflecting the potential COVID-19 transmissibility in the WHO European Region during the period of NPI relaxation after the first wave (Fig. A1).”.

Table A2. The basic reproduction number of SARS-CoV-2 across countries.

Country	Highest Rt*	First wave R0**	Country	Highest Rt*	First wave R0**
Austria	1.6	2.88	Iceland	2.12	-
Belgium	1.58	2.87	Israel	1.35	3.19
Bulgaria	1.63	2.66	Italy	1.71	2.77
Switzerland	2	2.42	Liechtenstein	1.91	-
Cyprus	1.96	-	Lithuania	1.7	2.19
Czechia	1.64	2.69	Luxembourg	1.76	3.23
Germany	1.49	2.35	Latvia	1.81	2.26
Denmark	1.69	1.74	Netherlands	1.66	2.05
Spain	1.57	3.3	Norway	1.7	2.05
Estonia	1.89	1.86	Poland	1.73	2.36
Finland	1.38	2.48	Portugal	1.46	2.5
France	1.48	2.64	Slovakia	1.57	2.77
United Kingdom	2.05	2.99	Slovenia	1.72	1.78
Croatia	1.75	2.54	Sweden	1.81	1.82
Hungary	2.34	2.8	Ukraine	1.24	4.6
Ireland	1.62	2.09			

*Highest Rt: the highest Rt estimated by Arroyo-Marioli et al. (2021) between 1 August 2020 to 1 December 2020.

**First wave R0: the estimates (Lai et al., 2021) of the basic reproduction number in the first wave of COVID-19, the absent country-specific estimates are setted as blank.

- 2. Communication improvements, adding missing information, model verification :

- It is difficult to understand the model and assess the suitability of the used model. A combination of high model complexity (different degrees of pooling at various parts of the model), missing information, and confusing presentation, makes it hard to understand the model. How to improve:

Reply: Many thanks for your detailed suggestions which significantly improved this manuscript.

- ****Improve presentation****:

- 2a) include a model graph that highlights the various sub-models and the degree of pooling used (see above).

- 2b) Also, the model is currently partially explained in different places. E.g. the fact that you fit a separate model for each country and month is only explained in the results, not in the methods. I recommend focussing on writing a good, concise (!), and comprehensive model description in the methods.

Reply: Thanks for your comment. We have concisely and comprehensively described our model in **Method, Assessing the effectiveness of NPIs and vaccination**. To give the readers a clear understanding of our model, we added a model graph in **Methods, Assessing the effectiveness of NPIs and vaccinations** according to your suggestions.

Fig. 5 Overview of models using bottom-up approaches. Orange nodes represent the observed data. Blue nodes represent the pseudo variables generated by the observed data. For each country, we put a prior on R_0 with hyperprior varying by country, where the prior mean was setted as the highest R_t before 1 December 2020, see Supplementary Information A2. Then, $R_{0,t}$ representing the intrinsic transmissibility was estimated by Model 1. By comparing observed R_t with $R_{0,t}$ in Model 2, we estimated coefficients of variables to estimate relative effects of various interventions and factors on COVID-19 for each country by month. A variable, represented by the residual Δ , was used to characterise the impact of other unknown factors on R_t in addition to practical vaccination rate, NPIs and air temperature. Finally, the overall effectiveness of NPIs and vaccination in the European region was evaluated in Model 3 by pooling the national effectiveness across countries through meta-analysis with the random-effect model.

- 2c) ****Correct mistakes and remove unclarities:****
- Take, for example, the equation between line 448 and 449:
 - First, this equation is wrong, as you say yourself in line 453. In fact, there is output noise on R_t , which is not included in this equation.

Reply: Sorry for the unclarity here. R_t we used was collected from *Our World in Data* which is estimated by Arroyo-Marioli et al., (2021). Thus the output noise on R_t we showed in line 453 in the previous version was associated with their model and has been processed by their estimator of R_t . We added a clear statement before the equation about output noise on R_t to show that R_t was collected from a publicly available dataset.

For your easy reference, we pasted the added statements here: *In this study, the daily estimates of R_t were obtained from the Our World in Data data repository and contributed by Arroyo-Marioli et al⁴⁴, with R_t being estimated from the number of daily new cases using the Kalman filter⁴⁵.*

- also, I assume that x, y, and z should likely be x_t, y_t, and z_t. Issues like this make it harder to understand the model.

- What about epsilon? Is there a different epsilon for each day (i.e. you should write epsilon_t), or only one per month? This is pretty crucial for understanding the model

- ****Add missing information:****

- 2d) This is a Bayesian analysis, but most priors are not given. E.g. what were the priors for alpha and beta?

- 2e) Especially, what was the prior for epsilon and how was it chosen? The prior on epsilon determines how much of the changes in R_t the model will explain with NPIs/vaccines, and how much it will attribute as to noise.

- (if you don't have already, you should probably have a hyperprior over epsilon)

Reply: Thanks for your comments above. We have carefully checked and revised symbols to make them consistent with our model. In **Assessing the effectiveness of NPIs and vaccinations**, the generalised linear model you referred to here has been revised, and we have added all the prior information over the coefficients of our model as below. For your easy reference, we pasted the added information here:

For month l, we built the following generalised linear model to use the variation of NPIs and vaccination explaining the reductions in R_t over time.

$$R_t^c = R_{0,t}^c \exp(-\alpha_l^c N_t^c - \beta_l^c V_t^c - \lambda_l^c N_t^c V_t^c - \varphi_l^c T_t^c - \Delta_l^c) = \Phi_{t,l}^c \quad (5)$$

where N_t^c , V_t^c , and T_t^c are the stringency index of NPIs, practical vaccination rate, and air temperature for country c in month l at day t, respectively. In addition to NPIs and vaccination, we also modelled their interaction of reducing R_t by directly incorporating a product term ($N_t^c V_t^c$) in our model. Moreover, the unobserved confounders of the change between $R_{0,t}^c$ and R_t^c were represented by the residual Δ_l^c . To estimate the model parameters, we used a Bayesian framework to provide the estimates with prior knowledge. We assumed that $R_t^c \sim \text{gamma}(\Phi_{t,l}^c, 0.5)$. As NPIs and vaccination were likely to positively impact the trajectories of COVID-19, i.e., reducing $R_{0,t}$, we put a gamma prior with hyperprior over the coefficients of NPIs, vaccination and their interaction term in our model. Specifically, α_l^c , β_l^c and λ_l^c , following $\text{gamma}(u, 1)$ and $u \sim \text{uniform}(0, 1)$, varied by country according to their data contexts. Additionally, we had a Gaussian prior over the coefficients $\varphi_l^c \sim N(0, 0.5)$ and $\Delta_l^c \sim N(0, 0.5)$, as temperature and other unknown factors might also be related to the transmission dynamics of the disease^{50,51}. The posterior estimates of coefficients in Eq. (5) can be found in Supplementary Information B3. Finally, the relative effectiveness of NPIs and vaccination for country c in month l could be calculated by $1 - \exp(-\alpha_l^c N_l^c)$ and $1 - \exp(-\beta_l^c V_l^c)$, respectively, wherein N_l^c and V_l^c was the average value of the stringency index of NPIs and practical vaccination rate. The effect size was defined as the reduction in $R_{0,t}$ regarding R_t , i.e., $1 - R_t/R_{0,t}$, with the combined effect of two independent variables calculated as the sum of the estimated effects of the two variables over the corresponding product of their effectiveness. The prior and posterior predictive estimations can be found in Supplementary Information C1.

- ****Add missing analysis for model verification****

- 2f) There are no prior or posterior predictive checks. I expect these in every Bayesian modelling paper (in the Supplement). How am I supposed to evaluate your model if I can't see the model fit?

Reply: Thanks for your comment. We have added both prior and posterior predictive checks for France (fully vaccination rate 70%), Israel (66%), Croatia (43%) and Bulgaria (21%) in the Supplementary Information C1. The results showed that posterior has more predictive accuracy than the prior.

Fig. C1 Comparison between our prior and posterior over the coefficients in our model for their predictability. Here, we take four countries with varied vaccination coverage (France 70%, Israel 66%, Croatia 43% and Bulgaria 21%) as examples. We first used our assumed prior to produce estimates of R_t by our model for the four countries, and then used the fitted model with posterior to generate another set of R_t for comparison.

- 2g) Ideally, you'd also do leave-one-out cross validation (or other forms of cross-validation) where you show the predictions of your model on unseen data. I don't think that the NPI stringency index and the "practical vaccination rate" alone can explain much of the changes in R_t , so I expect most of the fit to be carried by epsilon, and accordingly I expect the model to have very little predictive power.

Reply: Thanks for your comment. We validated our model by leave-one-out cross validation for each of the 31 study countries where the median RMSE was 0.27 (IQR: 0.23 – 0.33) and R-square ranged from 0.36 (Poland with a vaccination rate of 52%) to 0.75 (Ukraine with a vaccination rate of 16%) (Supplementary Information C4). We also changed our model settings and parameter selections to assess our model robustness. The output showed that the overall trends in estimates were highly consistent across experimental conditions (Supplementary Information B5).

In addition to the NPIs and vaccination, we also involved air temperature and a random variable (Δ_t^f) representing the unobserved confounders of the change between $R_{0,t}^c$ and R_t^c in Eq. (5). Overall, our model can explain the variation trend of reductions in $R_{0,t}$. While the unexplained sharp fluctuations may be caused by the variation of other factors, e.g., health resources and uncoordinated interventions, etc. However, these factors are hard to measure with our limited data, and the aim of this study is to explore the relative effectiveness of NPIs and vaccination in the real-world instead.

Fig. C4 The results of leave-one-out-validation over 31 study countries generated by pooling the national results. The overall R-square is 0.53.

Fig. C5 Comparison between the predicted real-time R_t and the empirical real-time R_t by pooling the national results. The predicted R_t were represented by green dots, where the empirical R_t were represented by red dots. Here, we take four countries with varied vaccination coverage (France 70%, Israel 66%, Croatia 43% and Bulgaria 21%) as examples.

- 2h) Epsilon is pretty crucial in your analysis. Please show epsilon over time (together with alpha and beta over time). Also show gamma.

Reply: The results for all the coefficients were given in the **B3. Estimates of the coefficients** section of Supplementary Information.

- ****Other missing information****

- 2i) Show the results for coefficients: All plots that claim to show effectiveness show $1 - \exp(-\alpha_i * x)$. In my mind, the effectiveness of NPIs and vaccines is only $1 - \exp(-\alpha)$, while x represent how strict/high the NPIs/vaccination rates were. $1 - \exp(-\alpha_i * x)$ should not be called "effectiveness", but rather something like "realised reduction in R_t ", or so. But regardless of terminology, you should show alpha and beta over time. The abstracts makes strong claims that you show how the effectiveness of NPIs develops as vaccination rates increase. But, in fact, Figure 2 might only show that fewer NPIs were used as vaccination rates increase.

Reply: Thanks for your comment. Yes, in previous similar studies (Flaxman et al., 2020; Brauner et al., 2021), the effectiveness is defined as $1 - \exp(-\alpha)$. This is because their NPIs are specific policies represented by nominal variables. Under the situation, when the policy exists, the policy variable equals to 1. However, we used stringency index which is a continuous variable ranged from 0 to 100 (normalised to 0 - 1 in the modelling), thus we kept x in $1 - \exp(-\alpha_i * x)$. We gave the estimates of all the coefficients in our model in the **B3. Estimates of the coefficients** section of Supplementary Information. Also, more details about stringency index can be found in Supplementary Information D2 section.

- 2j) Your dataset on vaccine efficiency by subtype: It is great that you collected this dataset. But please include some information about how you collected it. Also, Table A2, which shows the dataset, has no citations/sources what-o-ever. That is very suboptimal if the dataset is supposed to be a contribution.

Reply: Sorry for the missing data sources, we have added relevant citations for the Table, which is Table A3 in the revised version of Supplementary Information A3. We also made some modifications on calculating practical vaccination rate, please see our response to your following comment 3d).

- 3) specific issues:

- 3a) why negative binomial distribution on R_t (line 453)? R_t is a continuous variable, the negative binomial distribution is a discrete distribution.

Reply: Sorry for the wrong description. R_t follows gamma distribution as $R_t^c \sim \text{gamma}(\Phi_{t,l}^c, 0.5)$, where $\Phi_{t,l}^c$ is the generalised model we built in Eq. 5.

- 3b) The English is not perfect, but understandable. Even abstract has sentences that I don't understand, e.g. The effectiveness of NPIs alone declined approximately 23% since the introduction of vaccination strategies, where the relaxation of NPIs promoted the decline from May 2021. There are also some incorrect words used here and there. The paper would benefit from language editing

Reply: Proof reading to improve the language has been conducted by native English speakers.

- 3c) How did you select the countries?

Reply: Thanks for your comment. We first designed the model and then included 31 countries that have publicly available data for our analysis in the WHO European region.

- 3d) You use the "national daily practical vaccination rate" as this input variable. This is a misnomer, as this number is already calculated from vaccine efficiency numbers, not only vaccination rates. But besides, this number should already pretty much give you the reduction in R_t from vaccines at each point in time, so it's curious to use this as the model input. At the very least, you should compare the "national daily practical vaccination rate" against the reduction in R_t estimated by your model, and show if they are similar or not (and if not, explain why not)

Reply: Our study countries used different vaccine products in their mass vaccination. However, these vaccines might have different efficacy. Thus, to estimate a general effectiveness of vaccination, we need to standardise the input variable of vaccination, that is the full vaccination rate. And we used "practical vaccination rate" as a baseline to account for the different vaccines efficacy. We have further explained this in the first paragraph in Results and in the **Vaccination data** section of Methods. Nonetheless, we adopted your suggestion and only account for the different vaccine efficacy against initial SARS-CoV-2 strains rather than VOCs (see Supplementary Information A3) in the revised version, as we have already considered variants in the calculation of $R_{0,t}$.

In fact, Fig. 3a has given the comparison between the "national daily practical vaccination rate" against the reduction in R_t estimated by our model. The effectiveness of vaccination alone was plotted by the boxplot for different practical vaccination rate levels. It showed that the reduction in R_t estimated by our model was not the daily practical vaccination rate itself (even not a linear relationship). A possible reason is that we pooled national results to a European effect, where the regional effect has accounted for the different levels of vaccination progress across countries and time.

- 3e) question on the results: how is it possible that e.g. both NPIs and vaccines reduce R by ~25% in Sept 2021, but R overall is reduced by > 75% (see figure 2 bottom)

- In general, why are these reductions so small? R was mostly around 1 in the these periods, how did that happen if not through NPIs and vaccines? Probably epsilon in your model?

Reply: Many thanks for your comments. We believed that such a big gap was caused by the inappropriate large R_0 we setted in the previous version. As the reviewer suggested above, the population behaviour changed massively in the latter half of 2020 even without NPIs and vaccinations in place, the basic reproduction number for our analysis should be lower than that in the first wave. We consequently revised our R_0 according to the estimates of R_t (Arroyo-Marioli et al., (2021)) to avoid the influence of such behaviour change. The detailed revision can be found in Supplementary Information A2.

Besides, the change of trajectories of Covid-19 was very complex where NPIs and vaccination were not all driving factors. In the revised manuscript, we also provided the effect of air temperature and the unobserved confounders (unknown factors) in Supplementary Information B1. We found that NPIs associated with vaccination could reduce $R_{0,t}$ by 49%. Air temperature seems to have a small effect on reducing $R_{0,t}$ by -1% to 6%. And there is still a share of reductions in $R_{0,t}$ contributed by the unknown factors ranging from 9% to 21% during our study period.

- 3f) The vaccination effects that you get, even when factoring in that not the whole population is vaccinated, seem pretty inconsistent with results from clinical trials (which show higher effectiveness). Do you have any idea why?

Reply: The environment of clinical trials is relatively ideal, and many actual situations are not considered. The efficacy of Covid-19 vaccines is commonly evaluated in the short term, and the dynamics across time are unclear, e.g., not all the population completed their vaccination together but in a sequence way. The effectiveness of vaccination to prevent transmission among the whole population might also be different with the effectiveness indicator of vaccines to protect the infection at a susceptible individual level. In addition, our effectiveness of vaccination here is relative to the NPI. If the strength of the NPI increases, the effectiveness of the vaccine may also decrease. However, we are targeting the entire population to control the epidemic. When the vaccination rate is high enough to reach herd immunity, and the NPIs are released, the estimated effect might be very close to the actual effectiveness of the vaccine.

- 4) minor points:

- abstract should give countries and time window of analysis

Reply: Thanks for your comment. We have revised our abstract to give countries and the time window of analysis.

To address this, we built a Bayesian inference model to explore the changing effectiveness of NPIs and vaccination, based on a large-scale dataset including epidemiological parameters, virus variants, vaccines, and climate factors, for 31 countries in Europe from 1 August 2020 to 25 October 2021.

- Quote: "However, clinical trials estimating vaccine efficacy 61 were conducted when novel variants, such as the Delta-variant, had not yet emerged7,"

- not longer true now

Reply: The sentence has been removed.

- the intro could be shortened. I did not count words, but it feels long.

Reply : We have shortened the Introduction with 734 words in total now.

- how are explanatory variables normalised? what is the range e.g. of stringency index

Reply: All the explanatory variables were normalised by min-max normalisation. The original data of the stringency index we collected from Our world in Data (Mathieu et al., 2021) ranged from 0 to 100, where 0 represents no intervention and 100 represents the strictest interventions. After normalisation, all the explanatory variables in the model ranged from 0 to 1. In the revised manuscript, we added a description of how we normalised the explanatory data at the end of **Data sources and processing**.

In the modelling, all explanatory variables were normalised by min-max normalisation, ranging from 0 to 1.

Reviewer #3 (Remarks to the Author):

The authors used an epidemiological model to estimate empirical effectiveness of NPIs in the presence of vaccination and how vaccines affect the effectiveness of NPIs in 27 European countries. They found that NPIs are still important to suppress the virus transmission despite the administrations of vaccines in these European countries, which agrees with other recently published work (Sonabend R. 2021; Leung K. 2021). They also found that the administration of vaccines may reduce the effects of NPIs on reducing the virus transmission, while further clarifications about the methods are needed to assess this conclusion.

Reply: Thanks for your comments. We have referenced these two articles in appropriate positions.

One advantage of this work is the way that the authors dealt with the heterogenous transmissibility of the changing viral variants and various effectiveness of different vaccines across different countries. This could be helpful for future work that hopes to synthesise evidence from such complex real-world situations. However, I would be particular interested to know why the effectiveness of NPIs and vaccinations look almost identical to the observed stringency index of NPIs and vaccination rate. This is critical to interpret the findings as well.

Reply: The effect of interventions is defined as $1 - \exp(-\alpha_i * x)$ for both NPIs and vaccination. With respect to NPIs, the independent variable x is the observed stringency index of NPIs, and for vaccination, x is the practical vaccination rate we estimated from the reported vaccination rate and vaccine efficacy of vaccines. Therefore, the variation of effectiveness of NPIs and vaccination would look similar to the change of observed NPI stringency index and vaccination rate over time. However, we found that the relative contributions of NPIs and vaccination on reducing reproduction numbers were different. The impact of vaccination has exceeded that of NPIs on the trajectory of Covid-19 transmission in Europe since

August 2021. Although the role of vaccination had been flourishing since January 2021, we saw little growth in the relative effectiveness of Covid-19 in September - October 2021 during the circulation of the VOC Delta. We also added Fig. 3d to compare the practical vaccination rate with the corresponding effectiveness estimated by our model.

The authors studied an important and timely question, which could potentially be useful in informing the policy making. However, the fact that the authors used the stringency index to measure NPIs may limit the usefulness of their findings, as the stringency index is not very interpretable and people are still unsure about which NPIs should be implemented or relaxed. In addition, the authors adopted transmission (measured by reproduction number) as outcome instead of disease burden (i.e., severe outcomes after infection), which may further limit the implications of their findings as the disease burden may be substantially reduced by the vaccines while the virus is still transmitting. These limitations should also be discussed when interpreting their findings.

Reply: Thanks for our comment. As asynchronous and uncoordinated NPIs have been implemented across the world (Ruktanonchai et al., 2020, Hale et al., 2021), our modelling work used integrated NPI stringency index generated by the OxCGRT, which allows us to model and compare the effect of NPIs across countries. However, as pointed out by the reviewer, this index would bring some difficulties to interpret our findings. For specific countries, to use our results, they may need to refer to the OxCGRT's definition and deploy appropriate specific interventions based on their local context, such as culture and lifestyles, etc. In addition, we agree with the reviewer that it is important to understand the impact of different interventions on the Covid-19 related disease burden (severe illness, hospitalisation and death). However, this study was designed to quantify the effect of NPIs and vaccination on mitigating the transmission, and due to the complexity of disease burden and driving factors, it might be worthwhile to conduct a separate study for this in the future. We have acknowledged these limitations in the Discussion section.

First, as we focused on the effect of NPIs and vaccination in preventing COVID-19 transmission, this study did not investigate the impact of these interventions to reduce severe outcomes, e.g. hospitalisations and deaths, which warrants investigation in future studies. Fifth, randomised control trials cannot be performed to robustly examine causality between interventions and the reduction in COVID-19 transmission, and this study was not designed to distinguish the efficiency of individual NPIs and their interaction.

The writing of the manuscript could be further improved. Particularly, figures need to be appropriately referenced, while more details are needed in the methods. In addition, statements in the introduction and discussion could be more cautious.

Specific comments:

* I would suggest the authors to include page number and line number in the manuscript, so that it would be easier to locate.

Reply: Sorry for the inconvenience, we have added page number and line number in the revised manuscript.

Introduction

* “By mid-September 2021, the vaccination rate had reached 59.6% ...”: Are these numbers for partial or complete vaccination?

Reply: These numbers refer to complete vaccination. We have revised the sentence to give a clearer definition as: *For example, Europe reported a 7% increase in new weekly cases and 11% increase in COVID-19 attributed deaths during the week of 4 to 10 October 2021, compared with the previous week¹⁴, despite 59.6% of the population in the European Region having been fully vaccinated by mid-September 2021⁷.*

* “Despite these vaccination rates, a subsequent wave of Covid-19 cases emerged in July 2021 with daily confirmed cases of nearly 40,000, driven primarily by the emergence of the novel Delta variant.” This statement implies the decreased effectiveness of vaccines on preventing infections from the Delta variant, which may be true. In particular, the wave can be caused by infections among unvaccinated people, therefore it needs support from references that reported the vaccination status among the confirmed cases during this July 2021 wave.

Reply: Due to the length of Introduction, this paragraph has been rewritten and supported by relevant references. Now the relevant sentence reads: *However, the number of confirmed new COVID-19 cases across the world in 2021 remained high, and subsequent waves of transmission have occurred with emergence of more transmissible variants of concern (VOCs), e.g. Alpha and Delta due to immune evasion^{10,11} and the potential for reinfection amongst previously infected or vaccinated populations^{12,13}.*

* “...as to the true threshold of herd immunity” Do the authors mean the threshold of vaccination rate to reach herd immunity?

Reply: The sentence has been removed.

* “...epidemiological model-based numerical simulations...”: It does not read like a typical term in the field.

Reply: The sentence has been revised as “*Previous modelling studies^{17,18,19,20} have preliminarily explored the implementation and effectiveness of NPIs in the COVID-19 vaccination era.*”

Results

* The description of instantaneous basic reproduction number, instantaneous reproduction number and the estimation of effectiveness of NPIs and vaccines can be further improved. Currently, it is not very clear how these key parameters were defined and estimated without further reading the methods and supplements.

Reply: Thanks for your suggestions. In the revised **Results**, we briefly described the definition of instantaneous basic reproduction number, instantaneous reproduction

number and the estimation of effectiveness of NPIs and vaccines. Due to the length of the main text, the details of these parameters have been provided in the Methods and Supplementary Information.

* "... effectively immune via vaccination" - Do the authors mean the overall effectiveness of vaccines against variants? Perhaps need some clarification here.

Reply: Yes, we used to mean that. We have updated the Results, while this sentence has been removed in the revised manuscript.

The respective effectiveness of NPIs and vaccination over time (upper panel in Figure 2) looks almost identical with the observations of these two variables in Figure 1b. Did the authors estimate the overall effect or effect per unit of NPIs and vaccinations? If overall effect, the changes in the effectiveness may just because the changes in the stringency of NPIs and vaccination rates. If effect per unit was estimated, it is interesting to investigate why the effects changed in the same way with that of the data. It is not very clear how did the authors estimate the effectiveness based on the descriptions in results and methods.

Reply: We estimated the effect per unit of NPIs and vaccinations by estimating the coefficients of NPIs and vaccination. Finally, the relative effectiveness of NPIs and vaccination for country c in month l can be calculated by $1 - \exp(-\alpha_l^c N_l^c)$ and $1 - \exp(-\beta_l^c V_l^c)$, respectively, wherein N_l^c and V_l^c was the average value of the stringency index of NPIs and practical vaccination rate. The results illustrated in the main text were the pooling effect generated by the national results through meta-analysis. In the revised manuscript, we also provided the estimates of all coefficients across countries and over time, see Supplementary Information B3.

* "Before the onset of vaccination, NPIs alone controlled the practical transmissibility, measured by, to about 1.07 (1.00 - 1.15) together with the unobserved confounders but still larger than 1." - It's not clear which figure this sentence refers to and hard to follow.

Reply: The relevant content has been moved to Supplementary Information A2, while this sentence has been removed.

* "While the combination of NPIs and vaccination has decreased a larger share of during this period than before together with the same unobserved confounders." - The sentence is confusing.

Reply: The sentence has been removed.

* "The stringency index of NPIs should maintain 60 currently, while it can relax to 44 in the post-vaccination era." It is hard to interoperate which NPIs should be implemented or relaxed so that we could maintain the stringency index as 60 or 44.

Reply: Thanks for our comment. As countries implemented diverse NPIs packages without coordination (Ruktanonchai et al., 2020), we used government interventions' integrated stringency index generated by the OxCGRT as a proxy to estimate the general restraint of 'lockdown style' NPIs. The detailed definition is given in

Supplementary Information D2. Due to the diversity of interventions in different countries, here we can only indicate the suitable strength for their considered specific NPIs. Therefore, for specific countries, they may need to refer to the OxCGRT's approach that was used to generate the stringency index, and further deploy the appropriate specific interventions based on their local context, such as culture, lifestyles, etc. An example of how to use our analysis was given in Results, Potential relaxation of NPIs amid vaccination.

Nonetheless, we have added this as a limitation in Discussion. For your easy reference, we pasted the added words below:

Fifth, randomised control trials cannot be performed to robustly examine causality between interventions and the reduction in COVID-19 transmission, and this study was not designed to distinguish the efficiency of individual NPIs and their interaction.

* “openness risk” - worth a brief introduction as it's not a very well-known term. Also need to mention this in the methods.

Reply: Thanks for your comment. We have added a brief description of the openness risk directly after its first appearance, by “*The openness risk is a case-evidenced index of risk rating, related to whether a country is ready to adopt an ‘open’ policy (remove/reduce NPI measures).*”. We also added the description in Methods and Supplementary.

* “We found that our forecasted variations on NPIs implementation to stop Covid-19 were highly correlated.”

— 0.49 does not look like a very high correlation, though the test is significant.

Reply: Thanks for your comment. We have removed the sentence.

— It is interesting to note that there are a group of countries in the bottom-left (UK, Ireland, Portugal, Spain and Iceland), which seemed very different from the rest in the top-right. Actually, predictions for the vast majority of countries in the top-right look like negatively correlated with the openness risk, which is the opposite to the reported. What are the differences for those countries in the bottom-left?

Reply: We re-thought and discussed the content shown in Fig. 4b, and have added more discussion about the comparison between our “predictions” and the openness risk.

Fig. 4b shows that findings from these two indexes are generally consistent. For instance, countries falling in Group 1 should consider delaying relaxation or boosting their NPIs, and countries of Group 2 could consider relaxing their NPIs.

Discussion

* “Where the synergistic effect of NPIs and vaccination was 46.9% in September 2021.” Not sure which plot this sentence refers to.

Reply: This sentence refers to Fig. 2, which has been referenced in the revised manuscript. As we updated our data to 25 October 2021 and further modified our model according to reviewers' comments, the sentence has been revised as “*The effect*

of vaccination on preventing population-wide COVID-19 transmission gradually increased, and surpassed that of NPIs in August 2021 (Fig. 2). However, in the context of circulation of more transmissible variants, e.g. VOC Delta and Omicron^{26,27}, NPIs remain an important complementary to vaccination in reducing COVID-19 transmission before herd immunity has been reached, at least in the short term, when the combined effect of NPIs and vaccination only reached 49% in October 2021.”

* “Implementing NPIs with higher strength, such as restrictions of gatherings of more than 10 people compared to that of more than 1000 people, would further decrease the potential contact population of infections. Under the circumstance, the susceptible population is harder to contact the infections and become new infections then, if the probability of getting infected after contact is unchanged. Our estimates of NPIs effectiveness are consistent with these findings, and we also provide the effectiveness of NPIs over time to counter the influence of policy fatigue.” Not sure how did the authors conclude these. I don’t think findings from this study support this as no individual NPIs were investigated.

Reply: Thanks for this comment. This study used integrated NPI stringency index (SI) as a proxy to estimate the general restraint of ‘lockdown style’ interventions. We assumed that a higher SI would lead to a higher effectiveness on mitigating Covid-19 transmission, and the overall effect of NPIs was measured by $1 - \exp(-a \cdot SI)$. To better interpret our findings and avoid confusions as mentioned by the reviewer, this part has been revised as

“Additionally, more stringent NPIs, such as contact reductions and travel restrictions, can further increase their effects to decrease the transmission risk of the virus.”

* “...facing aggressive variants such as the Delta variant, over 80% of people need to have immunity to achieve herd immunity.” Assuming 100% effectiveness of protection from the vaccines? It is very unlikely to be true.

Reply: The sentence has been revised as *“Furthermore, limited by the weakened effect of various vaccine products against different variants and the delays of vaccine development and distributions, achieving herd immunity may be a big challenge, particularly in the face of highly transmissible variants such as Omicron or even Delmicron³⁸ (Supplementary Information D3).”*

* “The very population attacked by the recent outbreak of Covid-19, caused by the variants Delta, in China was children instead of previous young people [35], because most adults have been fully vaccinated. It evidenced the importance of the continued implementation of NPIs.” I agree that children may become more susceptible to infection due to their lower vaccination rate compared to the adults (although fully vaccinated does not mean no infection). However, I doubt if the cited reference can actually support that the Delta variant more likely to hit children. That the Fujian outbreak hit the children was also because the outbreak happened in a school setting, which was initiated by a student who was infected by the parent. At the same time, another Delta-related outbreak in Yangzhou hit mostly old adults initially, as the outbreak was seeded in two mah-jong places.

Reply: Thanks for your comment. To avoid confusion, this content has been removed in the revised manuscript.

* "..., unless they take precautions such as getting vaccinated and wearing masks." Not sure if I would fully agree with this statement. People can get infected with full vaccinations and mask wearing (Martín-Sánchez 2021). Consider to revise to something like "...due to the uncertainty of duration of protection."

Reply: Thanks. We have revised the sentence accordingly.

Methods

* SI Table A2 - suggest to provide references for these estimates.

Reply: Done.

* First equation in "Assessing the effectiveness of NPIs and vaccinations" section is confusing.

— Does the model fit to each county separately or together? If together, need to indicate the country in the notions.

Reply: Sorry for the unclarity. We fitted each country and each month separately, and then pooled the national results by meta-analysis by month. We have revised the section to give a clear and concise description of our model, with consistent symbols.

— Do the dependent variable (x, y, z) change over time and country?

— How do x and y correlate with the estimates in the previous sections (e.g., daily protection rate)?

Reply: Yes, we have revised our symbol system to make them consistent with our model. We have revised the equation as

For month l , we built the following generalised linear model to use the variation of NPIs and vaccination explaining the reductions in R_t over time.

$$R_t^c = R_{0,t}^c \exp(-\alpha_l^c N_t^c - \beta_l^c V_t^c - \lambda_l^c N_t^c V_t^c - \varphi_l^c T_t^c - \Delta_l^c) = \Phi_{t,l}^c \quad (5)$$

where N_t^c , V_t^c , and T_t^c are the stringency index of NPIs, practical vaccination rate, and air temperature for country c in month l at day t , respectively. In addition to NPIs and vaccination, we also modelled their interaction of reducing R_t by directly incorporating a product term ($N_t^c V_t^c$) in our model. Moreover, the unobserved confounders of the change between $R_{0,t}^c$ and R_t^c were represented by the residual Δ_l^c .

* Justify why hard norm distribution was used.

Reply: The reason of using half normal over the diffusion parameter (variance) of R_t^c is we assumed that $R_t^c \sim \text{gamma}(\Phi_{t,l}^c, \sigma)$, where $\sigma \sim \text{half_normal}(0, 0.5)$. As variance should always be a positive value, we used half normal to restrict $\sigma > 0$. In the revised manuscript, we directly assigned $\sigma = 0.5$, and conducted sensitivity analysis about the value, see Supplementary Information B5.

* Suggest to provide equation for the generalised additive regression.

Reply: The generalised additive regression is just the equation in “Assessing the effectiveness of NPIs and vaccinations” section. We have removed this sentence to avoid the confusion. In the revised manuscript, we directly introduced the equation as

For month l , we built the following generalised linear model to use the variation of NPIs and vaccination explaining the reductions in R_t over time.

$$R_t^c = R_{0,t}^c \exp(-\alpha_l^c N_t^c - \beta_l^c V_t^c - \lambda_l^c N_t^c V_t^c - \varphi_l^c T_t^c - \Delta_l^c) = \Phi_{t,l}^c \quad (5)$$

* “Finally, the impact of vaccination on the effectiveness of NPIs was defined by the difference between NPIs effect before and after the vaccination onset.”

— Need more details of how the difference was calculated. Is it absolute additive or ratio? An equation may be helpful.

Reply: We have revised the Methods section to give a clearer description of our model. The sentence has been removed.

— An important assumption here is that the authors assumed all changes in effectiveness of NPIs were caused by vaccinations, while other confounders do exist (like fatigue). Need to discuss this as a limitation.

Reply: Thanks for your comment. We have revised our model by directly adding an interaction term between NPIs and vaccination to model their interactions. And we also added the following discussion of the limitation, “*Third, vaccines might also be administered to people who were already infected, and the effectiveness of NPIs might be negatively reduced by policy fatigue or positively impacted by adherence to personal protective behaviours against COVID-19 infections. However, their impact has not been analysed in this work due to a lack of relevant data.*”

* MCMC - no description about the estimated parameters, prior distributions, likelihood functions, algorithms, iterations and diagnosis.

Reply: Sorry for the missing information. We added relevant settings before Fig. 5. For your easy reference, we pasted the added words here

As NPIs and vaccination were likely to positively impact the trajectories of COVID-19, i.e., reducing $R_{0,t}$, we put a gamma prior with hyperprior over the coefficients of NPIs, vaccination and their interaction term in our model. Specifically, α_l^c , β_l^c and λ_l^c , following $\text{gamma}(u,1)$ and $u \sim \text{uniform}(0,1)$, varied by country according to their data contexts. Additionally, we had a Gaussian prior over the coefficients $\varphi_l^c \sim N(0,0.5)$ and $\Delta_l^c \sim N(0,0.5)$, as temperature and other unknown factors might also be related to the transmission dynamics of the disease^{50,51}. The posterior estimates of coefficients in Eq. (5) can be found in Supplementary Information B3. Finally, the relative effectiveness of NPIs and vaccination for country c in month l could be calculated by $1 - \exp(-\alpha_l^c N_t^c)$ and $1 - \exp(-\beta_l^c V_t^c)$, respectively, wherein N_t^c and V_t^c was the average value of the stringency index of NPIs and practical vaccination rate. The effect size was defined as the reduction in $R_{0,t}$ regarding R_t , i.e., $1 - R_t/R_{0,t}$, with the combined effect of two independent variables calculated as the sum of the estimated effects of the two variables over the corresponding product of their effectiveness. The prior and posterior predictive estimations can be found in Supplementary Information C1.

We estimated the effect of NPIs and vaccination for every month to account for seasonal and other potential temporal effects. This process was performed using Markov Chain Monte Carlo (MCMC) methods with Rstan⁵². We ran 4 chains for 2000 iterations with 500 iterations of warmup and a thinning factor of 1 to obtain 600 posterior samples for each month and country (see Supplementary Information C2). We validated our model using a ‘leave-one-out’ cross-validation approach (see Supplementary Information C4). Sensitivity analyses were also performed to assess model robustness in terms of our assumptions on parameters (see Supplementary Information B5).

* Meta-analysis - It is not clear which figures are results from this analysis.

Reply: We have revised the relevant sentence to directly indicate which figures are results from this analysis. The revised sentence is

We pooled national effectiveness across the 31 study countries (Supplementary Information B2) to estimate the regional effect (Fig. 2 - 3) through meta-analysis using a random-effects model⁵³.

Figures

* Figure 1

— panel b - Does the “documented vaccination” indicate partial or full vaccination?
 — panel c - Does each column adds up to 1? Do panel c and d share the same color legend?

Reply: The “documented vaccination” indicates full vaccination. And each column does add up to 1 and panel c and d share the same colour legend. We have clearly defined these in the figure title.

* Figure 4:

— panel a - suggest to indicate the variable name and unit in the color legend.
 — panel b - not very clear what are the x- and y-axis based on the legend.

Reply: The missing information has been added and further clarified (see below).

Fig. 4 The possible relaxation of NPIs or the requirement of extra stringency to contain COVID-19 across countries. (a) Under the scenario of vaccination and COVID-19 transmission by 25 October 2021, required changes of NPI stringency index to contain COVID-19 ($R_t < 1$). The negative change means the degree of NPI relaxation, compared to the stringency on 25 October 2021. (b) The comparison between the estimated requirement of changes in NPI stringency index presented in (a) and the output of the openness risk (from 0 to 1) - an indicator modified from the OxCGRT's approach²⁵. A higher openness risk (>0.5) means an increasing likelihood of COVID-19 resurgence, and vice versa. Countries in Group 1 (increasing NPI stringency) and Group 2 (relaxing NPIs) mean that they have consistent findings between two indicators. Groups 3 and 4 mean that the two indicators have conflicting results and extra evidence might be needed.

References:

- Leung, Kathy, Joseph T. Wu, and Gabriel M. Leung. Effects of adjusting public health, travel, and social measures during the roll-out of COVID-19 vaccination: a modelling study. *The Lancet Public Health* 6.9 (2021): e674-e682.
- Martín-Sánchez, Mario, et al. COVID-19 transmission in Hong Kong despite universal masking. *Journal of Infection* (2021).
- Sonabend, R, et al. Non-pharmaceutical interventions, vaccination, and the SARS-CoV-2 delta variant in England: a mathematical modelling study. *The Lancet* (2021).

Reply: These references have been cited in the revised manuscript.

Reference in this response letter

Brauner, J.M., Mindermann, S., Sharma, M., Johnston, D., Salvatier, J., Gavenčiak, T., Stephenson, A.B., Leech, G., Altman, G., Mikulik, V. and Norman, A.J., 2021. Inferring the effectiveness of government interventions against COVID-19. *Science*, 371(6531).

Arroyo-Marioli F, Bullano F, Kucinskas S, Rondón-Moreno C (2021) Tracking R of COVID-19: A new real-time estimation using the Kalman filter. *PLoS ONE* 16(1): e0244474. <https://doi.org/10.1371/journal.pone.0244474>

Lai, S., Ruktanonchai, N.W., Carioli, A., Ruktanonchai, C.W., Floyd, J.R., Prosper, O., Zhang, C., Du, X., Yang, W. and Tatem, A.J., 2021. Assessing the Effect of Global Travel and Contact Restrictions on Mitigating the COVID-19 Pandemic. *Engineering*.

Sharma, M., Mindermann, S., Rogers-Smith, C., Leech, G., Snodin, B., Ahuja, J., Sandbrink, J.B., Monrad, J.T., Altman, G., Dhaliwal, G. and Finnveden, L., 2021. Understanding the effectiveness of government interventions against the resurgence of COVID-19 in Europe. *Nature communications*, 12(1), pp.1-13.

Flaxman, S., Mishra, S., Gandy, A., Unwin, H.J.T., Mellan, T.A., Coupland, H., Whittaker, C., Zhu, H., Berah, T., Eaton, J.W. and Monod, M., 2020. Estimating the effects of non-pharmaceutical interventions on COVID-19 in Europe. *Nature*, 584(7820), pp.257-261.

Brauner, J.M., Mindermann, S., Sharma, M., Johnston, D., Salvatier, J., Gavenčiak, T., Stephenson, A.B., Leech, G., Altman, G., Mikulik, V. and Norman, A.J., 2021. Inferring the effectiveness of government interventions against COVID-19. *Science*, 371(6531).

Mathieu, E., Ritchie, H., Ortiz-Ospina, E. et al. A global database of COVID-19 vaccinations. *Nat Hum Behav* (2021). <https://doi.org/10.1038/s41562-021-01122-8>

Ruktanonchai, N.W., Floyd, J.R., Lai, S., Ruktanonchai, C.W., Sadilek, A., Rente-Lourenco, P., Ben, X., Carioli, A., Gwinn, J., Steele, J.E. and Prosper, O., 2020. Assessing the impact of coordinated COVID-19 exit strategies across Europe. *Science*, 369(6510), pp.1465-1470.

Hale, T., Angrist, N., Goldszmidt, R., Kira, B., Petherick, A., Phillips, T., Webster, S., Cameron-Blake, E., Hallas, L., Majumdar, S. and Tatlow, H., 2021. A global panel database of pandemic policies (Oxford COVID-19 Government Response Tracker). *Nature Human Behaviour*, 5(4), pp.529-538.

Wang, S.Y., Juthani, P.V., Borges, K.A., Shallow, M.K., Gupta, A., Price, C., Won, C.H. and Chun, H.J., 2021. Severe breakthrough COVID-19 cases in the SARS-CoV-2 delta (B. 1.617. 2) variant era. *The Lancet Microbe*.

Stan Development Team (2020). RStan: the R interface to Stan. R package version 2.21.2. <http://mc-stan.org/>.

REVIEWER COMMENTS

Reviewer #2 (Remarks to the Author):

- The authors addressed several of my points, the new model figure is nice! I'm now also convinced that it's acceptable to fit an individual model to each country/month and then use meta-analysis; although I still think that a full model with fixed pooling would be preferable. There remain several issues, see below. Overall, my tentative view is that the paper is likely not quite strong enough for Nature Communications, and might fit better with a subjournal. But it's a borderline decision, which the editor will have to take.

- Major issues:

- The method description is still not very clear. A major thing that I can't figure out: Do you run joint inference across every country, per month (i.e. one inference/model per month)? Or do you run a separate inference for each country in each month (i.e. one model/inference for each pair of country/month, i.e. 31x more inference runs than the other option). From the text it looks like you're doing the latter. But if so, then the hyperpriors over alpha, beta, and lambda don't make any sense. These are pointless if you fit a separate model for each country and month.

Hyperpriors/partial pooling only make sense if you do joint inference! In the current model, they only obfuscate things and should be removed (and the priors adapted accordingly). Note that the hyperprior on $R_{0,t}$ is not pointless, as you do joint inference over the days in a month.

- Figure B11 shows that the unknown factor variance matters a lot for the results (S5 and S6), but the noise scale is not chosen in any principled way. If you do joint inference over the countries in each month, you should just put a hyperprior over the unknown factor variance. If you fit a single model for each country in each month, then your options are more limited. You could choose the noise scale by cross-validation, or you could at least plot the prior predictive for a few values and explain why you choose 0.5. Currently, no reason is given why you choose 0.5.

- There is still a major communication problem! "Effectiveness" is not the same as "reduction in R_t ". Take, for example, Figure 2. The y-axis label says "delta R_t (%)". The y-axis is correct, as far as I can tell. But the figure caption says "effectiveness of NPIs and vaccinations"; and this is also how the results are presented in the abstract. These quantities are obviously not identical. E.g. assume that a country uses has only vaccinated a small percentage of the population at a point in time. Then delta R_t (%) from vaccinations would be very small. But that doesn't mean that vaccines are not effective! The author need to be clear and precise in their terminology. Also make sure you use "effect" vs "effectiveness" consistently. You also need to be very clear in what your key messages are. The results section reads pretty fluffy at the moment.

- prior predictive checks: As far as I can tell, you show one draw from the prior. This is not enough for evaluating the model, you need to show the prior predictive distribution; or at least many draws from the prior. Additionally, it looks like the prior is not reasonable, as it extends to unrealistically high values of R .

- Minor

- Figure B6: something is wrong with the vaccination coefficients. Early on, then there are no vaccinations, the vaccination coefficient should revert to the prior. However, the prior does not have a mean of 0, it should have a mean of roughly 0.5! ($\text{Gamma}(0.5,1)$)

- in the main text, they say they use the *data* from Arroyo-Marioli et al. In the Supplement A, they say they used the *methodology* from Arroyo-Marioli et al.. Needs to be clarified!

- In my original review, I asked this: "3f) The vaccination effects that you get, even when factoring in that not the whole population is vaccinated, seem pretty inconsistent with results from clinical trials (which show higher effectiveness). Do you have any idea why?" - The authors had a good answer to this question in the response letter, make sure it is also in the paper.

- Methods: Capital Phi is not explained anywhere. Also, either equation 5 or line 471 is wrong. Either $R_t = \text{PHI}$, or $R_t \sim \text{gamma}(\text{PHI}, 0.5)$. Both at the same time is not possible.

Reviewer #3 (Remarks to the Author):

It is appreciated that the authors have addressed most of my concerns. There is one remaining point that, I think, needs to be further addressed and clarified.

My remaining concern is about the interpretation of the relative effectiveness of NPIs and vaccinations. According to the authors' clarifications, the per unit coefficients (shown in Supplementary Figure B5 and B6) were estimated for NPIs stringency and vaccination rate, respectively, which were then multiplied by the corresponding independent variables to derive the total effects (shown in Figure 2A). It looks like the per unit coefficient for NPIs were actually pretty stable during the study period, even after the vaccination were introduced (Supplementary Figure B5), while the total effects of NPIs (Figure 2A) decreased in a pattern that reflects changes in NPIs stringency (Figure 1B). As such, the effectiveness of NPIs were actually not affected by the introduction of vaccination, but affected by the changes in NPIs stringency.

The current phrase of results and indications in the manuscript sounds like the NPIs were not as effective as it was before vaccinations were introduced (e.g., line 45-46 in abstract), which I don't think is what the results indicate. Given that the per unit effectiveness of NPIs did not substantially change over time, you could also expect a decreased total effects of NPIs for a lower NPIs stringency even before the vaccinations were introduced. Therefore, it is important to rephrase the results and indications to make this point clear in the manuscript.

POINT-BY-POINT RESPONSE TO REVIEWER COMMENTS

Reviewer #2 (Remarks to the Author):

- The authors addressed several of my points, the new model figure is nice! I'm now also convinced that it's acceptable to fit an individual model to each country/month and then use meta-analysis; although I still think that a full model with fixed pooling would be preferable. There remain several issues, see below. Overall, my tentative view is that the paper is likely not quite strong enough for *Nature Communications*, and might fit better with a subjournal. But it's a borderline decision, which the editor will have to take.

Reply: We are very grateful to the reviewer's insightful comments for improving our manuscript. Considering the following reasons, we believe this manuscript fits well within the focus of *Nature Communications*, and will be of broad and high interest to readers. First, it is of importance to understand the real-world impact of vaccination, NPIs and their interactions on Covid-19 transmission across countries and over time, under various stringency of interventions. The findings of our study could provide important insights for tailoring the Covid-19 interventions in the vaccination era, not only in Europe, but also in countries that are still implementing strict NPIs in the mass vaccination. Second, the modelling framework used in this study is a typical approach, i.e., the Bayesian inference model, which has been used in studies published in leading journals such as *Nature*, *Science*, and *Nature Communications*, to assess the effectiveness of NPIs against Covid-19 in the early stages of the pandemic (Flaxman et al., 2020; Brauner et al., 2021; Sharma et al., 2021). In addition, the comments from reviewers for improving our methodologies have made our study more solid and robust.

As suggested by the reviewer, in previous revision, we already conducted a full model with fixed pooling as a comparison (Supplementary Information C4). However, compared with the full model with fixed pooling across all countries, overall, these two pooling approaches have present similar patterns, but the meta-analysis approach for pooling the national results has a better explanation power (16% increase) regarding r-square (SI Fig. C4 and C6). This might be because the meta-based pooling approach could better explain the heterogeneous data context in local situation for each country (SI Fig. B4), although the full model with fixed pooling could help understand the overall effects of interventions across all countries. We have included these analyses and results in the Supplementary Information.

- Major issues:

- The method description is still not very clear. A major thing that I can't figure out: Do you run joint inference across every country, per month (i.e. one inference/model

per month)? Or do you run a separate inference for each country in each month (i.e. one model/inference for each pair of country/month, i.e. 31x more inference runs than the other option). From the text it looks like you're doing the latter. But if so, then the hyperpriors over alpha, beta, and lambda don't make any sense. These are pointless if you fit a separate model for each country and month. Hyperpriors/partial pooling only make sense if you do joint inference! In the current model, they only obfuscate things and should be removed (and the priors adapted accordingly). Note that the hyperprior on $R_{0,t}$ is not pointless, as you do joint inference over the days in a month.

Reply: Thanks for these comments. We did not fit a separate model for each pair of country and month, because, as the reviewer said, it doesn't make any sense. We conducted joint inference for each country to include all months, i.e., one inference per country. We ran joint inference over the months in every country to determine the mean for the corresponding month. Therefore, hyperpriors over alpha, beta, and lambda were included, as we assumed that the relative impact of NPIs or vaccination, and other factors (e.g. policy fatigue and changing human behaviours) on COVID-19 might vary across time within each country. In the revised manuscript, we further clarify that *"We used a bottom-up approach (described in Fig. 5) to evaluate the effect of NPIs and vaccination in Europe by pooling the national effect through meta-analysis. For each country c , we fitted a Bayesian model across all months by assuming that the effect of NPIs and vaccination on reducing COVID-19 transmission was relatively stable and constant in each month l ."*

- Figure B11 shows that the unknown factor variance matters a lot for the results (S5 and S6), but the noise scale is not chosen in any principled way. If you do joint inference over the countries in each month, you should just put a hyperprior over the unknown factor variance. If you fit a single model for each country in each month, then your options are more limited. You could choose the noise scale by cross-validation, or you could at least plot the prior predictive for a few values and explain why you choose 0.5. Currently, no reason is given why you choose 0.5.

Reply: We fitted a single model over months for each country. Fig. B11 (now Fig B12) shows that the variance of unknown factor might not significantly change the results over time for each country. The border trends across different settings were consistent. Thus, we chose the middle value as the default setting of our model. In this revision, based on your suggestions, we tested the prior predictive for different values. We chose 0.5 because it produced lowest RMES compared to the other values. We added this to Supplementary Information as the reason why we choose 0.5 on page 26.

- There is still a major communication problem! "Effectiveness" is not the same as "reduction in R_t ". Take, for example, Figure 2. The y-axis label says "delta R_t (%)". The y-axis is correct, as far as I can tell. But the figure caption says "effectiveness of NPIs and vaccinations"; and this is also how the results are presented in the abstract. These quantities are obviously not identical. E.g. assume that a country uses has only

vaccinated a small percentage of the population at a point in time. Then delta R_t (%) from vaccinations would be very small. But that doesn't mean that vaccines are not effective! The author need to be clear and precise in their terminology. Also make sure you use "effect" vs "effectiveness" consistently. You also need to be very clear in what your key messages are. The results section reads pretty fluffy at the moment.

Reply: The term 'effectiveness' of interventions in the reduction of R_t among populations has been widely used in previous publications for assessing NPIs (e.g. Flaxman et al., 2020; Brauner et al., 2021; Sharma et al., 2021). To be consistent, previously we used this term and gave a clear definition of effectiveness in lines 178-180: "*The effectiveness of NPIs and vaccination, thus, was defined as the contributed percentage reductions in reproduction number from $R_{0,t}$ to R_t , denoted as $\Delta R_t(\%) = 1 - R_t/R_{0,t}$ (see Methods).*" However, regarding concerns raised by the reviewer, we have replaced 'effectiveness' with 'reduction in R_t ' or 'effect of NPIs on reducing R_t /COVID-19 transmission' in this revision, to avoid any confusion caused by the use of the term that also refers to how effectively vaccines can prevent individual infections in clinical trials or in the real world. Besides, we restructured the Results section to make it clear and clean.

- prior predictive checks: As far as I can tell, you show one draw from the prior. This is not enough for evaluating the model, you need to show the prior predictive distribution; or at least many draws from the prior. Additionally, it looks like the prior is not reasonable, as it extends to unrealistically high values of R.

Reply: In this revision, we sampled 1000 values before calculating R values and gave the corresponding mean values. In the previous version, the unrealistically high R values are because we only produced one sample from the prior, where the small-probability values might be used in the predictive check. The Figure of revised prior predictive checks (see below) has been added in the Supplementary Information Fig. C1.

- Minor

- Figure B6: something is wrong with the vaccination coefficients. Early on, then there are no vaccinations, the vaccination coefficient should revert to the prior. However, the prior does not have a mean of 0, it should have a mean of roughly 0.5! (Gamma(0.5,1))

Reply: Sorry for the confusion. We put the vaccination coefficient to be 0 when there are no vaccinations. We have corrected this and given the practical values around 0.8. Noting that we put hyperprior over the mean of vaccination coefficients, and we run inference across month per country to generate the posterior mean coefficient for the corresponding country. Thus, when there are no vaccinations, its coefficient does revert to the prior of the uniform distribution from 0 to 1, but revert to the posterior mean value (around 0.8) inferred by all-months data for each country. The revised Figure B6 (now SI Figure B7) is provided below:

- in the main text, they say they use the *data* from Arroyo-Marioli et al. In the Supplement A, they say they used the *methodology* from Arroyo-Marioli et al.. Needs to be clarified!

Reply: Sorry for the confusion. We further clarified it in Supplementary Section A. The revised sentence reads “*In this study, the used instantaneous reproduction number, denoted as R_t , was estimated by Arroyo-Marioli et al.¹*”.

- In my original review, I asked this: "3f) The vaccination effects that you get, even when factoring in that not the whole population is vaccinated, seem pretty inconsistent with results from clinical trials (which show higher effectiveness). Do you have any idea why?" - The authors had a good answer to this question in the response letter, make sure it is also in the paper.

Reply: Thanks for the reminder. The relevant response has been added to the Discussion section on page 15.

- Methods: Capital Phi is not explained anywhere. Also, either equation 5 or line 471 is wrong. Either $R_t = \text{PHI}$, or $R_t \sim \text{gamma}(\text{PHI}, 0.5)$. Both at the same time is not possible.

Reply: Thanks for the comment. We have revised equation 5 as

$$R_t^c \sim \text{gamma}(\Phi_{t,l}^c, 0.5)$$

$$\Phi_{t,l}^c = R_{0,t}^c \exp(-\alpha_l^c N_t^c - \beta_l^c V_t^c - \lambda_l^c N_t^c V_t^c - \varphi_l^c T_t^c - \Delta_l^c)$$

Reviewer #3 (Remarks to the Author):

It is appreciated that the authors have addressed most of my concerns. There is one remaining point that, I think, needs to be further addressed and clarified.

My remaining concern is about the interpretation of the relative effectiveness of NPIs and vaccinations. According to the authors' clarifications, the per unit coefficients (shown in Supplementary Figure B5 and B6) were estimated for NPIs stringency and vaccination rate, respectively, which were then multiplied by the corresponding independent variables to derive the total effects (shown in Figure 2A). It looks like the per unit coefficient for NPIs were actually pretty stable during the study period, even after the vaccination were introduced (Supplementary Figure B5), while the total effects of NPIs (Figure 2A) decreased in a pattern that reflects changes in NPIs stringency (Figure 1B). As such, the effectiveness of NPIs were actually not affected by the introduction of vaccination, but affected by the changes in NPIs stringency.

Reply: Thank you for the comments. Our results indicated that the changes of the relative effect of NPIs came from two sources: one is the changes in NPI stringency, and the other is the interaction with vaccination. We agree that the per-unit coefficient for NPIs was pretty stable during the study period, which makes the total effects of NPIs (Figure 2A) decreased in a pattern that reflects changes in NPIs stringency (Figure 1B). However, the effect of NPIs with the same stringency was relatively reduced when vaccination rates increased (Figure 3B). One possible explanation for this is if people got fully vaccinated, they might have immunity to prevent infections, whether they adopted NPIs or not. And consequently, the implementation of NPIs with the same stringency might be less important in reducing COVID-19 transmission than that in previous waves before mass vaccination. We have added these into the Discussion.

The current phrase of results and indications in the manuscript sounds like the NPIs were not as effective as it was before vaccinations were introduced (e.g., line 45-46 in abstract), which I don't think is what the results indicate. Given that the per unit effectiveness of NPIs did not substantially change over time, you could also expect a decreased total effects of NPIs for a lower NPIs stringency even before the vaccinations were introduced. Therefore, it is important to rephrase the results and indications to make this point clear in the manuscript.

Reply: Agree with the reviewer. Theoretically, the efficacy of each individual NPI with the same stringency will be the same over time under ideal and controlled circumstances, e.g. clinical trials. However, as mentioned in our response to previous comments, the changes of the relative effect of NPIs might be co-affected by the NPI stringency and the interaction with vaccination. Although the per-unit coefficient for NPIs was stable across months, the reduced stringency would result in the decreased effects of NPIs. In addition, the relative importance of NPIs with the same stringency would also reduce after vaccines rolled out. These might explain why the overall

effect of NPIs reduced in recent waves, compared to the first (Brauner et al., 2021; Flaxman et al., 2020) and second COVID-19 waves (Sharma et al., 2021) in Europe. We have revised these in the Abstract and Discussion. Besides, to give a more clear and straightforward results section, we further rephrased and restructured the results section.

Reference:

1. Brauner, J.M., Mindermann, S., Sharma, M., Johnston, D., Salvatier, J., Gavenčiak, T., Stephenson, A.B., Leech, G., Altman, G., Mikulik, V. and Norman, A.J., 2021. Inferring the effectiveness of government interventions against COVID-19. *Science*, 371(6531).
2. Flaxman, S., Mishra, S., Gandy, A., Unwin, H.J.T., Mellan, T.A., Coupland, H., Whittaker, C., Zhu, H., Berah, T., Eaton, J.W. and Monod, M., 2020. Estimating the effects of non-pharmaceutical interventions on COVID-19 in Europe. *Nature*, 584(7820), pp.257-261.
3. Sharma, M., Mindermann, S., Rogers-Smith, C., Leech, G., Snodin, B., Ahuja, J., Sandbrink, J.B., Monrad, J.T., Altman, G., Dhaliwal, G. and Finnveden, L., 2021. Understanding the effectiveness of government interventions against the resurgence of COVID-19 in Europe. *Nature communications*, 12(1), pp.1-13.

Reviewers' comments:

Reviewer #2 (Remarks to the Author):

Some things are improved:

- The methods section is now clearer
- The communication problem around "effectiveness" vs "reduction in R" has been addressed

Some things have not improved much, and I thus have remaining concerns about the quality of the model.

- My concern about the lacking justification for the unknown factor variance of 0.5 is essentially not addressed. The authors say that they "chose the middle value", which is meaningless, given that all three tested values are arbitrary. And then, they justify the choice of 0.5 by RMSE from the mean of the prior predictive distribution (across the three arbitrary values). That is not a reasonable way to choose hyperparameters that I've ever heard of. I suggested 2 reasonable ways to choose the hyperparameter in my last comments (hyper prior or cross validation).
- Prior predictive: I asked the authors explicitly to show the prior predictive *distribution*. Instead, they show the mean of the prior predictive distribution. This is pretty useless as a prior predictive check. In my last comment, I highlighted unrealistically high values of R in the one sample of the prior predictive that the authors showed. Given that the authors again chose to not show the full prior predictive distribution, I have to assume that the prior predictive distribution probably indeed doesn't too good.

Other points

- From the abstract: "Our results demonstrate that NPIs were complementary to, or even synergistic with, vaccination in an effort to reduce COVID-19 transmission"
- Where is the synergy shown? AFAICT, Fig. 3b shows that NPIs actually less effective at higher vaccination rates (as expected)

Reviewer #3 (Remarks to the Author):

Thank you for clarifications in the manuscript and responses. I think now we are on the same page. It perhaps could make the analysis and implications more clear if the authors change "relative effects of NPIs (or vaccines)" to "effects attributed to NPIs (or vaccines)". Otherwise, I think this work is already good enough for publication.

POINT-BY-POINT RESPONSE TO REVIEWER COMMENTS

Reviewer #2 (Remarks to the Author):

Some things are improved:

- The methods section is now clearer
- The communication problem around "effectiveness" vs "reduction in R" has been addressed

Some things have not improved much, and I thus have remaining concerns about the quality of the model.

- My concern about the lacking justification for the unknown factor variance of 0.5 is essentially not addressed. The authors say that they "chose the middle value", which is meaningless, given that all three tested values are arbitrary. And then, they justify the choice of 0.5 by RMSE from the mean of the prior predictive distribution (across the three arbitrary values). That is not a reasonable way to choose hyperparameters that I've ever heard of. I suggested 2 reasonable ways to choose the hyperparameter in my last comments (hyper prior or cross validation).

Response: According to your suggestion, we have revised the unknown factor variance from 0.5 to a hyperprior of Half normal (0.3). The updated results (see revised Fig. 2 below) are consistent with our previous findings, and all relevant figures and tables have been updated. Furthermore, we found that this approach could slightly improve the model's predictivity, or explanatory ability in most countries, in terms of R-squared values (see revised Supplementary Table C1 below).

Fig. 2 The effects of NPIs and vaccination on reducing COVID-19 transmission

in Europe over time. The overall monthly effects of interventions on reducing $R_{0,t}$ across 31 countries from 1 August 2020 to 25 October 2021 are presented with mean and 95% CI, which was pooled from national level to regional level using meta-analysis. The total effect of NPIs presented here is the effect of NPIs alone plus their interaction effect with vaccination, and the total effect of vaccination shown is the impact of vaccination alone plus its interaction effect with NPIs.

Table C1. R-squared values for the leave-one-out cross validation across countries using different choices of unknown factor variance. * using a hyperprior of Half normal (0.3) **using an unknown factor variance of 0.5. Δ is the difference between the revised results and previous results of R-squared.

Country	R^2^*	R^2^{**}	Δ	Country	R^2^*	R^2^{**}	Δ
Austria	0.57	0.57	0	Liechtenstein	0.55	0.52	0.03
Belgium	0.61	0.61	0	Lithuania	0.37	0.38	-0.01
Bulgaria	0.43	0.38	0.05	Luxembourg	0.69	0.64	0.05
Cyprus	0.42	0.42	0	Latvia	0.40	0.38	0.02
Czechia	0.44	0.44	0	Netherlands	0.46	0.46	0
Croatia	0.44	0.38	0.06	Norway	0.46	0.47	-0.01
Denmark	0.46	0.47	-0.01	Poland	0.35	0.36	-0.01
Estonia	0.49	0.40	0.09	Portugal	0.70	0.69	0.01
Finland	0.64	0.68	-0.04	Switzerland	0.40	0.37	0.03
France	0.63	0.62	0.01	Spain	0.64	0.64	0
Germany	0.70	0.69	0.01	Slovakia	0.44	0.46	-0.02
Hungary	0.45	0.43	0.02	Slovenia	0.49	0.47	0.02
Ireland	0.60	0.60	0	Sweden	0.46	0.47	-0.01
Iceland	0.75	0.68	0.07	Ukraine	0.76	0.75	0.01
Israel	0.62	0.61	0.01	United Kingdom	0.58	0.57	0.01
Italy	0.61	0.59	0.02	Average	0.55	0.53	0.02

- Prior predictive: I asked the authors explicitly to show the prior predictive *distribution*. Instead, they show the mean of the prior predictive distribution. This is pretty useless as a prior predictive check. In my last comment, I highlighted unrealistically high values of R in the one sample of the prior predictive that the authors showed. Given that the authors again chose to not show the full prior predictive distribution, I have to assume that the prior predictive distribution probably indeed doesn't too good.

Response: Apologies for our misunderstanding of the reviewer's previous comment on how to present the prior predictive distribution. According to reviewer's suggestion in previous comments, we did draw many samples from our prior distribution to produce

estimates of R_t in our last revision. However, we were struggling to show the distribution for each and every country as there are 13,950 graphs (450 days *31 countries) in total, and it is hard to read the figures if we put all approximate prior predictive distributions over each day for each country. Now, the full prior and posterior predictive distributions for each country over each day are provided in a separate file uploaded to Google drive as the file is large (<https://drive.google.com/file/d/1q2WQyDi4Q8KESLpDXdDoRN9q8TG7yKst/view?usp=sharing>).

To summarize the findings, in this revision, we have also added the corresponding 95% confidence interval to the mean value of the distribution. Moreover, taking four countries with varied vaccination coverage (France 70%, Israel 66%, Croatia 43% and Bulgaria 21%) as examples, we also presented the detailed prior predictive distribution in the Supplementary Information Figs. C1-C9, with some figures shown below for your information. Noting that our prior selection is sensible as all the observed R_t fall in the prior distributions, and the posterior distributions can better capture the variation of the observed R_t due to the largely increased R-squared values (Fig. C1).

Fig. C1 Comparison between our prior and posterior over the coefficients in our model for their predictability. We only present the mean values and 95% confidence interval here.

Fig. C2 The prior predictive distribution in France. The time arrow line is from right to left and from bottom to top. The first day, 1/8/2020, is shown at the bottom right. The observed R_t values are illustrated by dots.

Fig. C3 The posterior predictive distribution in France. The time arrow line is from right to left and from bottom to top. The first day, 1/8/2020, is shown at the bottom right. The observed R_t values are illustrated by dots.

Other points

- From the abstract: "Our results demonstrate that NPIs were complementary to, or even synergistic with, vaccination in an effort to reduce COVID-19 transmission"

- Where is the synergy shown? AFAICT, Fig. 3b shows that NPIs actually less effective at higher vaccination rates (as expected)

Response: Yes, the effect attributed to NPIs became less at higher vaccination rates. However, Fig. 2 also shows the effect of combined NPIs and vaccination (in black) on reducing R_t was higher than NPIs or vaccination alone. Given that vaccines were gradually rolled out over time and cannot provide 100 percent protection, the implementation of NPIs can somewhat further prevent the transmission among populations. That is why we argued that NPIs could be synergistic with vaccination in reducing COVID-19 transmission. However, to avoid potential confusion, we have removed this word from the Abstract.

Reviewer #3 (Remarks to the Author):

Thank you for clarifications in the manuscript and responses. I think now we are on the same page. It perhaps could make the analysis and implications more clear if the authors change "relative effects of NPIs (or vaccines)" to "effects attributed to NPIs (or vaccines)". Otherwise, I think this work is already good enough for publication.

Response: Many thanks for your efforts to improve this manuscript. We have changed our expression from "relative effects of NPIs (or vaccines)" to "effects attributed to NPIs (or vaccines)" as suggested.

REVIEWERS' COMMENTS

Reviewer #2 (Remarks to the Author):

Thanks, the authors have convincingly addressed my remaining concerns now.